# $G_{12/13}$-mediated signaling stimulates hepatic glucose production and has a major impact on whole body glucose homeostasis

Srinivas Pittala [1] ✉, Dhanush Haspula[1], Yinghong Cui[1], Won-Mo Yang [2], Young-Bum Kim [2], Roger J. Davis [3], Allison Wing [4], Yaron Rotman [4], Owen P. McGuinness [5], Asuka Inoue [6,7] & Jürgen Wess [1] ✉

Altered hepatic glucose fluxes are critical during the pathogenesis of type 2 diabetes. G protein-coupled receptors represent important regulators of hepatic glucose production. Recent studies have shown that hepatocytes express GPCRs that can couple to $G_{12/13}$, a subfamily of heterotrimeric G proteins that has attracted relatively little attention in the past. Here we show, by analyzing several mutant mouse strains, that selective activation of hepatocyte $G_{12/13}$ signaling leads to pronounced hyperglycemia and that this effect involves the stimulation of the ROCK1-JNK signaling cascade. Using both mouse and human hepatocytes, we also show that activation of endogenous sphingosine-1-phosphate type 1 receptors strongly promotes glucose release in a $G_{12/13}$-dependent fashion. Studies with human liver samples indicate that hepatic *GNA12* (encoding $G\alpha_{12}$) expression levels positively correlate with indices of insulin resistance and impaired glucose homeostasis, consistent with a potential pathophysiological role of enhanced hepatic $G_{12/13}$ signaling.

Hepatic glucose fluxes play a central role in maintaining euglycemia[1-3]. When blood glucose levels rise after a meal, elevated insulin levels promote glucose storage in the liver in the form of glycogen. Under hypoglycemic conditions, glycogen is broken down into glucose through glycogenolysis and released into the blood stream[1-3]. In addition, hypoglycemia-induced glucagon release promotes hepatic gluconeogenesis, leading to enhanced hepatic glucose production (HGP). HGP is increased in type 2 diabetes (T2D), leading to elevated blood glucose levels, in particular under fasting conditions. A detailed understanding of the signaling pathways and molecules that modulate hepatic glucose metabolism is therefore of great potential translational relevance.

Like essentially all other cell types, hepatocytes express dozens of GPCRs on their cell surface[4,5]. GPCRs, upon binding of extracellular ligands, activate distinct classes of heterotrimeric G proteins, which are composed of four major classes, the $G_s$, $G_i$, $G_q$, and $G_{12/13}$ families[6]. Previous studies have shown that GPCRs that activate $G_s$, $G_i$, or $G_q$ signaling in hepatocytes play important roles in regulating HGP and maintaining euglycemia[5,7]. In contrast, the potential importance of $G_{12/13}$ in regulating hepatic glucose fluxes remains unexplored. Studies in this area have been hampered by the lack of $G_{12/13}$-specific inhibitors and the fact that GPCRs that exclusively activate $G_{12/13}$ have not been identified. $G\alpha_{12}$ and $G\alpha_{13}$ are ~70% identical in amino acid sequence, are expressed in virtually all cell types, and have similar but not identical functional properties[8-10]. Interestingly, recent studies have shown that more than 30 GPCRs are able to couple to $G_{12/13}$, in addition to other functional classes of heterotrimeric G proteins[10-13]. This class of receptors includes, for example, receptors for sphingosine-1-

[1]Molecular Signaling Section, Laboratory of Bioorganic Chemistry, NIDDK, NIH, Bethesda, MD, USA. [2]Division of Endocrinology, Diabetes, and Metabolism, Beth Israel Deaconess Medical Center and Harvard Medical School, Boston, MA, USA. [3]Program in Molecular Medicine, University of Massachusetts Chan Medical School, Worcester, MA, USA. [4]Liver & Energy Metabolism Section, Liver Diseases Branch, NIDDK, NIH, Bethesda, MD, USA. [5]Departments of Molecular Physiology and Biophysics, Vanderbilt University School of Medicine Basic Sciences, Nashville, TN, USA. [6]Graduate School of Pharmaceutical Sciences, Tohoku University, Sendai, Miyagi 980-8578, Japan. [7]Graduate School of Pharmaceutical Sciences, Kyoto University, Kyoto 606-8501, Japan. ✉e-mail: Srinivas.pittala@nih.gov; jurgenw@niddk.nih.gov

phosphate, lysophosphatidic acid, angiotensin II, as well as thrombin and other proteases[10–13].

Receptor-activated $G\alpha_{12/13}$ subunits are able to interact with and activate members of the RH domain-containing guanine nucleotide exchange factors for Rho (RH–RhoGEF) family of proteins, leading to the formation of the active form of RhoA (RhoA-GTP)[8,9]. The RH-RhoGEF family consists of p115RhoGEF, leukemia-associated RhoGEF (LARG), and PDZ-RhoGEF[14]. RhoA-GTP promotes the activation of Rho kinase (ROCK), the major downstream effector of the $G\alpha_{12/13}$ -RhoA signaling pathway. ROCK activation then leads to the phosphorylation of various substrates including myosin light chain (MLC) phosphatase, ezrin/radixin/moesin (ERM), LIM kinase (LIMK), and numerous other cellular proteins[14]. Receptor- mediated stimulation of $G_{12/13}$ signaling also leads to the activation of c-Jun N-terminal kinase (JNK), resulting in various effects on gene expression and cellular functions[15–18]. Additional studies have shown that signaling via $G_{12/13}$ modulates cell mobility, growth, differentiation, and various transcriptional processes[10,14,19].

Changes in the expression levels of $G\alpha_{12}$ and $G\alpha_{13}$ have been demonstrated in numerous human diseases, and accumulating evidence suggests that $G_{12/13}$-mediated cellular signaling contributes to various pathophysiological disorders[10]. For example, Kim et al.[20] showed that liver steatosis was exacerbated in mice lacking $G\alpha_{12}$ in hepatocytes. The same group also demonstrated that the expression levels of $G\alpha_{12}$[20] and $G\alpha_{13}$[21] were significantly reduced in patients with steatohepatitis (NASH) and T2D, respectively.

To explore the potential role of hepatic $G_{12/13}$ signaling in regulating glucose homeostasis, we took advantage of the recent development of a designer GPCR that selectively activates $G_{12/13}$ signaling following treatment with certain small synthetic compounds that are overwise pharmacologically inert[13,22]. Specifically, we generated a mouse line that selectively expressed this designer receptor in hepatocytes. In parallel, we also generated mice that selectively lacked both $G\alpha_{12}$ and $G\alpha_{13}$ in hepatocytes. Systematic metabolic phenotyping studies with these mutant mouse strains demonstrated that stimulation of hepatic $G_{12/13}$ signaling promotes HGP via a ROCK1/JNK-dependent pathway, resulting in impaired glucose homeostasis. These findings suggest that strategies aimed at inhibiting hepatic $G_{12/13}$ signaling may prove useful for the treatment of T2D and related metabolic disorders.

## Results

### Generation of hepatocyte-specific G12D mice

We recently reported the development of a designer GPCR (designer receptor exclusively activated by a designer drug; abbreviated as DREAAD) that selectively couples to $G_{12}$ (official name: M3D-GPR183/ICL3)[13]. Like other members of the DREADD family, this recently developed DREADD can be selectively activated by clozapine-N-oxide (CNO), a small molecule which is pharmacologically inert, at least when used in the proper dose or concentration range[23,24]. In a recent study[22], we demonstrated that the $G_{12}$ coupling selectivity of M3D-GPR183/ICL3 could be further improved by introducing the $F^{1.57}V$ point mutation (Ballesteros-Weinstein numbering system for GPCRs)[25]. By using an enhanced sensitivity version of the NanoBiT-G protein dissociation assay, we now show that the M3D-GPR183/ICL3-$F^{1.57}V$ receptor is able to couple to both $G_{12}$ and $G_{13}$ with a preference toward $G_{12}$ (Supplementary Fig. 1). For the sake of simplicity, we refer to this DREADD construct as G12D throughout this manuscript.

Functional studies showed that CNO treatment (1 and 10 μM) of G12D-expressing HEK293A cells had no significant effect on intracellular inositol monophosphate ($IP_1$) and cAMP levels (Supplementary Fig. 2). These data indicate that ligand stimulation of G12D does not lead to the activation of $G_{q/11}$ or $G_s$, respectively.

To generate mice that express G12D in a hepatocyte-specific fashion, we used a recently developed mouse strain (official name:

Rosa26-LSL-G12D-IRES-GFP)[22]. In this mouse strain, cell-type-specific expression of G12D is achieved by Cre recombinase[22], due to the presence of a "loxP-stop-loxP" (LSL) cassette preceding the G12D coding sequence (Fig. 1a). Throughout the manuscript, we refer to these genetically modified mice simply as *LSL-G12D* mice. To create hepatocyte-specific G12D mice (Hep-G12D mice), we injected *LSL-G12D* mice with an adeno-associated virus (AAV-TBG-Cre) in which the expression of Cre recombinase is under the transcriptional control of the hepatocyte-specific thyroxine-binding globulin (TBG) promoter[26]. For control purposes, we injected *LSL-G12D* mice with the AAV-TBG-eGFP virus which codes for a physiologically inert protein (eGFP). Both viruses were injected into the tail vein of *LSL-G12D* mice[26].

Two weeks after virus administration, we monitored hepatic G12D expression via Western blotting. To detect the expression of G12D receptor protein, we used a primary antibody that recognized the N-terminal hemagglutinin (HA) tag that had been fused to the N-terminus of G12D (Fig. 1a). As expected, no immunoreactive species were observed in lysates from the liver or other metabolically important tissues prepared from *LSL-G12D* mice injected with the control virus (Fig. 1b). In contrast, *LSL-G12D* mice that had received the AAV-TBG-Cre virus selectively expressed G12D in the liver (Fig. 1b). Western blotting studies confirmed that the G12D designer receptor was expressed only in hepatocytes and not in other cell types of the liver (Kupffer cells, stellate cells, etc.) prepared from Hep-G12D mice (Supplementary Fig. 3). In the following, we refer to *LSL-G12D* mice injected with AAV-TBG-Cre or AAV-TBG-eGFP simply as Hep-G12D or control mice, respectively.

To determine which percentage of hepatocytes expressed the G12D receptor, we used flow cytometry to detect the presence of the HA epitope tag present at the extracellular N-terminus of G12D (Fig. 1a) (see Methods for details). This analysis showed that $55 \pm 8\%$ of purified hepatocytes prepared from Hep-G12D mice expressed the G12D construct (mean ± s.e.m.; $n = 3$). In contrast, no significant HA signal was detected with hepatocytes derived from control littermates ($n = 3$).

We next used qRT-PCR technology to compare the hepatic expression levels of the G12D designer receptor with those of several GPCRs known to be endogenously expressed in mouse hepatocytes. Specifically, we determined transcript levels of the following GPCRs (gene names in parentheses): glucagon receptor (*Gcgr*), $V_{1A}$ vasopressin receptor (*Avpr1a*), sphingosine 1-phosphate receptor subtype 1 (*S1pr1*), and Gpr91 (*Sucnr1*). Using RNA prepared from Hep-G12D mice, we found that the G12D designer receptor was expressed at similar levels as the endogenously expressed glucagon receptor (Supplementary Fig. 4; note that lower $\Delta C_t$ values correspond to higher transcript levels). Moreover, hepatic *G12D* mRNA levels were only 2-4-fold higher than the corresponding *S1pr1* and *Sucnr1* transcript levels. These observations indicated that Hep-G12D mice express G12D at levels similar or close to those of GPCRs that are endogenously expressed by the liver. Absolute $C_t$ values were (means ± s.e.m.; $n = 3$): *G12D*, $18.7 \pm 0.1$; *Gcgr*: $18.5 \pm 0.1$; *Avpr1a*, $24.5 \pm 0.1$; *S1pr1*, $20.9 \pm 0.2$; and *Gpr91*, $19.8 \pm 0.1$.

### Activation of hepatocyte G12D signaling results in pronounced hyperglycemia in various in vivo metabolic tests

In the absence of an activating DREADD ligand, Hep-G12D mice and control littermates did not differ in body weight, and fed and fasting blood glucose and plasma insulin levels (Supplementary Fig. 5). To explore the potential effects of activating hepatic $G_{12/13}$ signaling on blood glucose levels, we treated control and Hep-G12D mice (males; regular chow diet) with a single dose of CNO (3 mg/kg, i.p.), followed by the measurement of blood glucose levels. Prior to CNO treatment, mice had either free access to food ('fed' mice) or had been subjected to an overnight fast (12 hr). Strikingly, CNO treatment of Hep-G12D mice, but not of control littermates, led to a pronounced increase in blood glucose levels in both fed and fasted mice ('CNO challenge test';

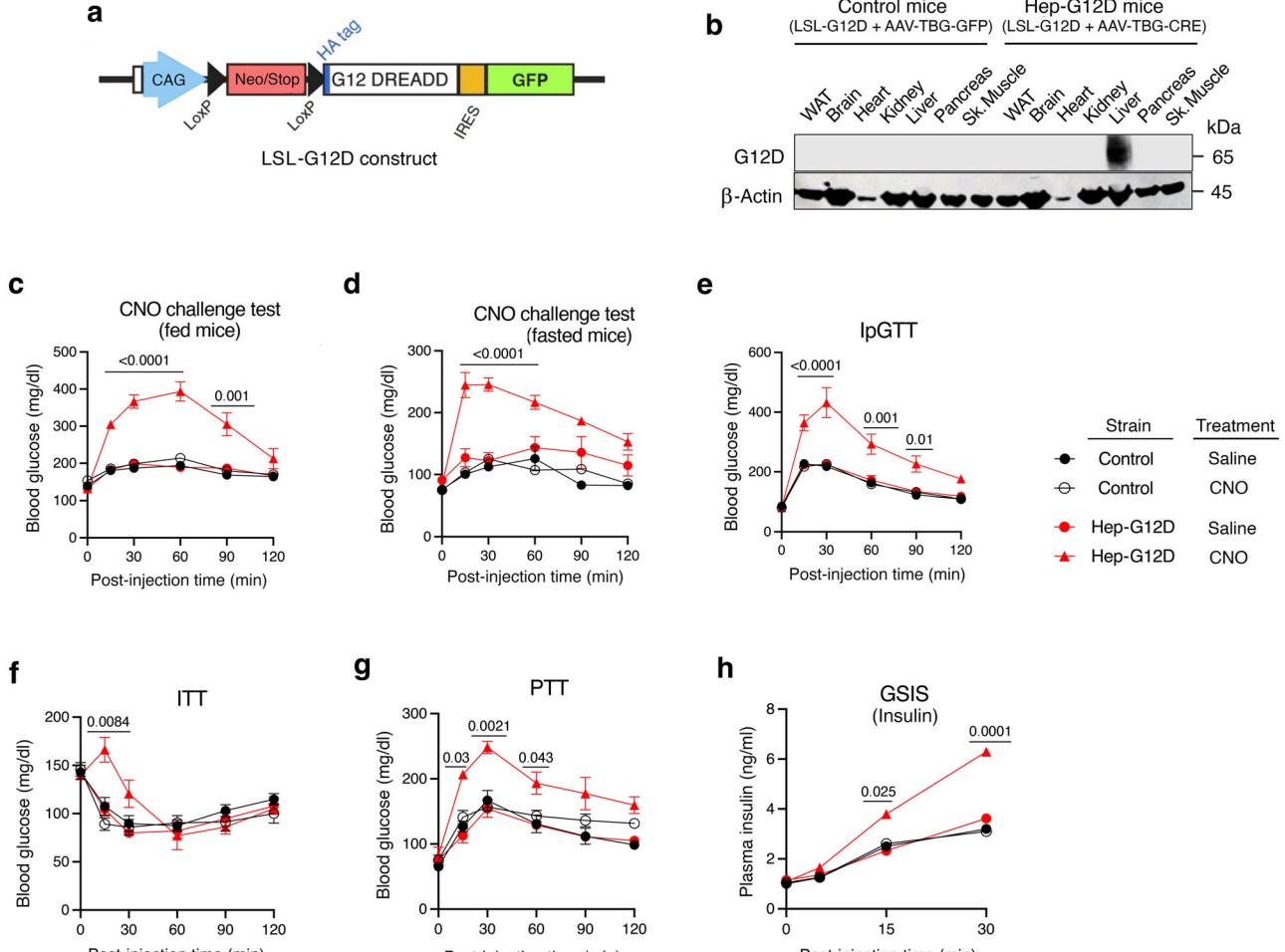

**Fig. 1 | In vivo metabolic studies with Hep-G12D mice maintained on regular chow. a** The indicated construct was inserted into the mouse *Rosa26* locus, resulting in LSL-G12D mice[22]. **b** Immunoblot showing selective expression of G12D in the liver of LSL-G12D mice following i.v. injection with the AAV-TBG-Cre virus (Hep-G12D mice). The G12D receptor was detected with an anti-HA antibody that recognized the HA epitope tag that had been fused to the N-terminus of G12D (**a**). G12D was not expressed in LSL-G12D mice treated with the AAV-TBG-eGFP control virus (control littermates). **c–h** In vivo metabolic tests performed with Hep-G12D mice and control littermates. **c, d** CNO challenge tests. Freely fed (**c**) or fasted (**d**) mice (12 h overnight fast) were injected i.p. with CNO (3 mg/kg) or saline, followed by monitoring of blood glucose levels (*n* = 8 per group). **e** I.p. glucose tolerance test

performed after a 12 h fast (IpGTT, 2 g glucose/kg) (*n* = 8 per group). **f** Insulin tolerance test carried out after a 4 h fast (ITT, 0.75 U insulin/kg, i.p.) (*n* = 8 per group). **g** Pyruvate tolerance test following a 12 hr fast (PTT, 1 g sodium pyruvate/kg, i.p.) (*n* = 8 per group). **h** Glucose-stimulated insulin secretion (GSIS; 2 g glucose/kg, i.p.). Following injection of the glucose bolus, plasma insulin levels were measured at the indicated time points (*n* = 8 per group). All studies were performed using 3-4-month-old male mice maintained on regular chow. Data represent means ± s.e.m. Numbers above horizontal bars refer to *p*-values. Statistical significance was determined by 2-way ANOVA followed by Bonferroni's post-hoc test (**c–h**). Sk. muscle, skeletal muscle. Source data are provided as a Source Data file.

Fig. 1c, d), suggesting that activation of hepatic $G_{12/13}$ signaling strongly promotes hepatic glucose production (HGP). Consistent with this notion, CNO-treated Hep-G12D mice showed significant impairments in glucose tolerance (i.p. glucose tolerance test; IpGTT; Fig. 1e) and insulin sensitivity (i.p. insulin tolerance test; ITT; Fig. 1f). Moreover, in a pyruvate challenge test (PTT), a test that is widely used to study in vivo gluconeogenesis, CNO-treated Hep-G12D mice displayed greatly enhanced blood glucose excursions, as compared to all other experimental groups (Fig. 1g). CNO treatment of Hep-G12D mice also led to a significant increase in glucose-stimulated insulin secretion (GSIS; Fig. 1h), most likely due to enhanced insulin release triggered by greatly elevated blood glucose levels (Fig. 1c, d). Consistent with the findings observed with male Hep-G12D mice, CNO (3 mg/kg, i.p.) treatment of female Hep-G12D mice also resulted in striking elevations in blood glucose levels in CNO challenge and glucose tolerance tests (Supplementary Fig. 6).

In a different set of experiments, we maintained Hep-G12D mice and control littermates on a calorie-rich high-fat diet (HFD) for at least

8 weeks to induce obesity, hyperglycemia, and other metabolic deficits[27]. CNO (3 mg/kg, i.p.) treatment of obese Hep-G12D mice also resulted in pronounced elevations of blood glucose levels in both fed and fasted mice (Supplementary Fig. 7), indicating that activation of hepatic $G_{12/13}$ signaling promoted striking hyperglycemic effects independent of whether mice were lean or obese. Hep-G12D mice and control littermates showed very similar body weights after consuming the HFD for 9 weeks (control mice: 49.7 ± 1.1 g; Hep-G12D mice: 49.8 ± 1.0 g; *n* = 8 per group; 17-week-old males).

## CNO treatment of Hep-G12D mice stimulates hepatic glycogenolysis and gluconeogenesis

To study hepatic glucose fluxes in greater detail, we carried out isotope labeling studies using chronically catheterized, conscious Hep-G12D mice and control littermates (males; see Methods for details)[28–30] (Fig. 2). Following a 5 hr fast, [6,6-D₂]glucose was infused continuously into the jugular vein to measure the glucose appearance (Ra). Total body water was enriched with D₂O, enabling us to calculate the

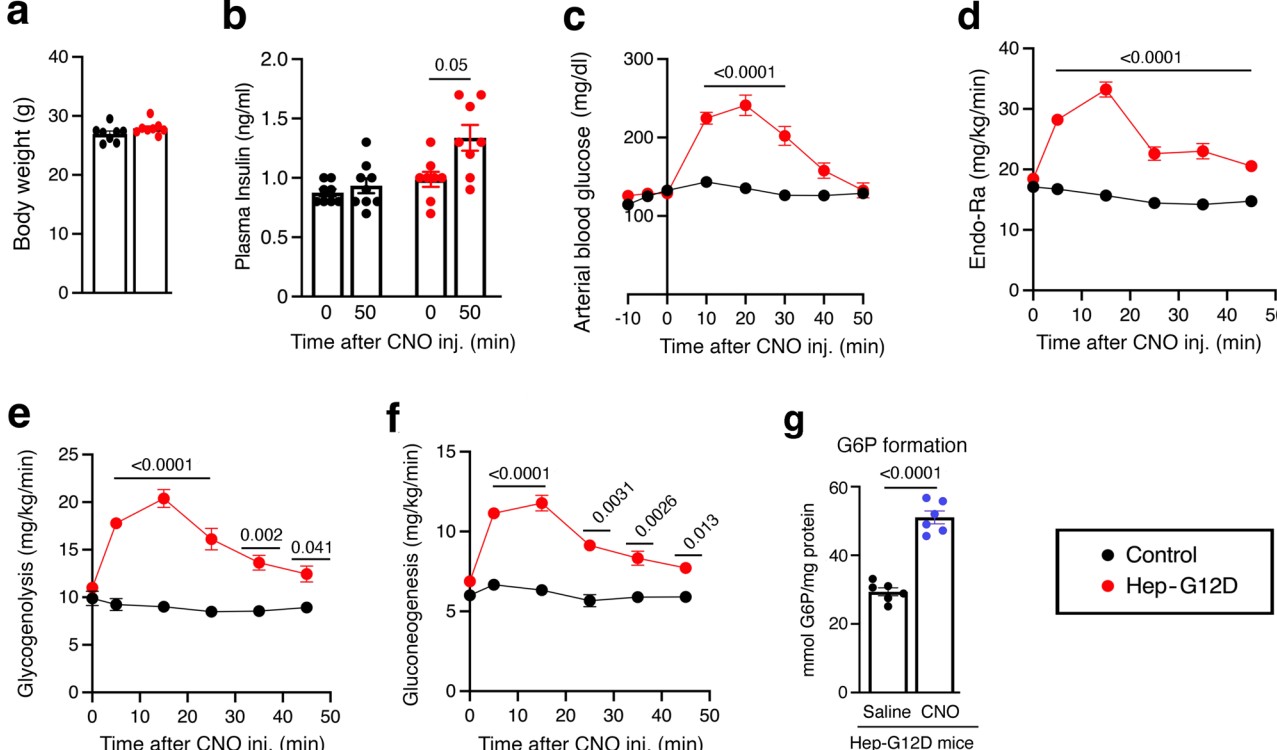

**Fig. 2 | In vivo euglycemic clamp studies with Hep-G12D mice.** Effect of CNO on hepatic glucose fluxes in Hep-G12D mice in vivo. All studies were carried out with male Hep-G12D mice and control littermates maintained on regular chow. **a** Body weight (age: 15 weeks; n = 8 per group). **b** Plasma insulin levels before and 50 min after CNO injection (n = 8 or 9 per group). **c–f** Changes in arterial blood glucose levels (**c**), rate of glucose appearance (endogenous glucose flux; Endo-Ra) (**d**), hepatic glycogenolysis (**e**), and gluconeogenesis (**f**), following treatment of Hep-G12D mice and control littermates with CNO (3 mg/kg, i.v.) (n = 8 or 9 per group).

All studies (**a–f**) were carried out with chronically catheterized, conscious 15-week-old male mice. **g** Hepatic glucose-6-phosphate (G6P) formation. Following a 4 h fast, Hep-G12D mice (12-week-old males) were treated with saline or CNO (3 mg/kg, i.v.). Five min later, livers were collected, and G6P levels were determined in liver lysates (n = 6 mice/group). Data represent means ± s.e.m. Numbers above horizontal bars refer to p-values (**a–f**, 2-way ANOVA followed by Bonferroni's post-hoc test; (**g**), two-tailed unpaired Student's t test). Source data are provided as a Source Data file.

contribution of gluconeogenesis and glycogenolysis to Ra[31]. Following assessment of baseline glucose enrichment, Hep-G12D and control littermates received an i.v. bolus of CNO (3 mg/kg). CNO treatment of Hep-G12D mice, but not of control littermates, resulted in robust and sustained increases in arterial glucose concentrations and Ra (Fig. 2c, d). The increase in Ra observed with CNO-treated Hep-G12D mice (Fig. 2d) was due to marked increases in the rates of both glycogenolysis and gluconeogenesis (Fig. 2e, f).

Glucose-6-phosphatase (G6Pase) catalyzes the final enzymatic step that promotes the conversion of G6P to glucose and it subsequent release after activation of both glycogenolysis and gluconeogenesis[2,32]. As shown in Fig. 2e, f, activation of hepatocyte G12D signaling in vivo resulted in a rapid increase in both of these processes. In agreement with this finding, CNO treatment (3 mg/kg, i.v.) of fasted Hep-G12D mice stimulated hepatic G6P formation within minutes (Fig. 2g). This observation is in agreement with the rapid onset of hepatic glucose production following CNO administration[32].

**CNO-induced hyperglycemia in Hep-G12D mice is due to hepatic G12/13 signaling**

We next wanted to confirm that the CNO-induced increases in blood glucose levels observed with CNO-treated Hep-G12D mice were indeed mediated by G proteins of the G12/13 family. To address this question, we initially injected wild-type (WT) C57BL/6 mice with the AAV-TBG-G12D virus, resulting in mice expressing G12D in the liver (Fig. 3a). CNO treatment (3 mg/kg, i.p.) of the AAV-TBG-G12D-injected WT mice led to robust increases in blood glucose levels (Fig. 3a), comparable in magnitude to those seen with CNO-treated Hep-G12D mice (Fig. 1c, d;

Supplementary Fig. 8). In contrast, CNO treatment of WT mice injected with the AAV-TBG-eGFP control virus showed only minor elevations in blood glucose levels (Fig. 3a), most probably caused by the injection stress.

We next generated a mouse line that expressed G12D selectively in hepatocytes, lacked Gα13 in this cell type, and did not express functional Gα12 throughout the body (abbreviated strain name: Hep-G12D G12/G13 KO mice). To generate this mouse strain, we co-injected (i.v.) *Gna12−/− Gna13 fl/fl* mice (genetic background: C57BL/6) with AAV-TBG-G12D and AAV-TBG-Cre (Fig. 3b, c). For control purposes, we co-treated *Gna12−/− Gna13 fl/fl* mice with AAV-TBG-G12D and AAV-TBG-eGFP (instead of AAV-TBG-Cre). To generate additional control mice, we treated age-matched WT mice with the same genetic background (C57BL/6) with two different virus mixtures, AAV-TBG-G12D plus AAV-TBG-Cre and AAV-TBG-eGFP plus AAV-TBG-Cre, respectively (Fig. 3b).

Consistent with the data shown in Fig. 3a, CNO-injected WT mice co-treated with AAV-TBG-G12D plus AAV-TBG-Cre resulted in robust hyperglycemic responses, in contrast to WT mice co-treated with AAV-TBG-eGFP plus AAV-TBG-Cre (Fig. 3b). The magnitude of this CNO response was significantly reduced in whole body Gα12 KO mice expressing G12D in hepatocytes (*Gna12−/− Gna13 fl/fl* mice co-treated with AAV-TBG-G12D plus AAV-TBG-eGFP) (Fig. 3b), suggesting that hepatic G12 signaling contributes to G12D-mediated hyperglycemia. Finally, G12D-mediated elevations in blood glucose levels were completely abolished in whole body Gα12 mice expressing G12D in hepatocytes but lacking Gα13 in this cell type ((*Gna12−/− Gna13 fl/fl* mice co-treated with AAV-TBG-G12D plus AAV-TBG-Cre) (Fig. 3b, c). This finding clearly indicates that the G12D-mediated hyperglycemic responses

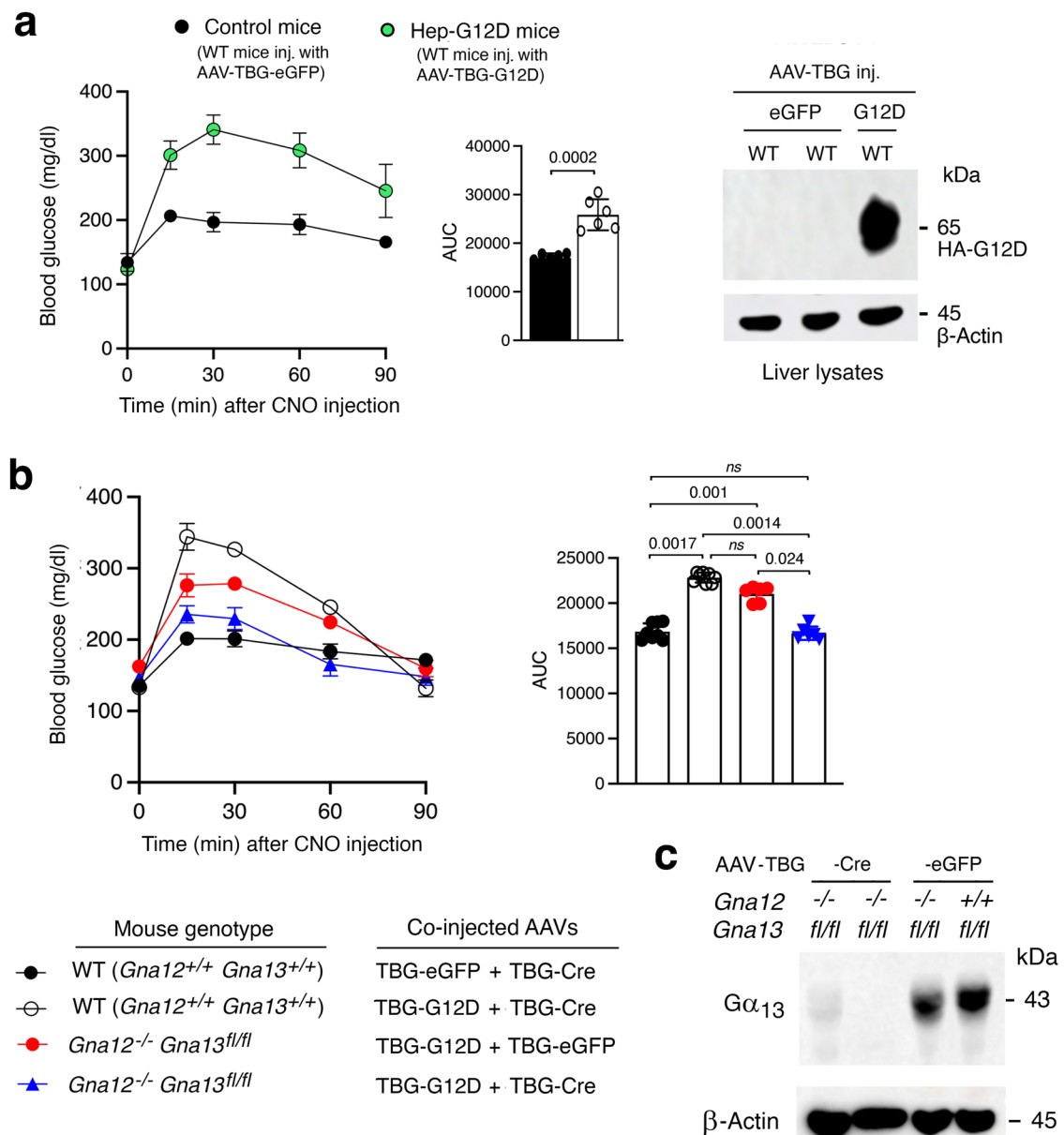

**Fig. 3 | G12D-mediated hyperglycemic effects require hepatic G$_{12/13}$ signaling.** WT mice (background: C57BL/6) and *Gna12−/−Gna13$^{fl/fl}$* mutant mice with the same genetic background were maintained on regular chow for 8 weeks. Subsequently, WT mice were injected i.v. with AAV-TBG-eGFP (control mice) or AAV-TBG-G12D (Hep-G12D mice). One week later, freely fed mice were injected i.p. with CNO (3 mg/kg) or saline, followed by the monitoring of blood glucose levels. **a** CNO treatment of WT mice injected with the G12D virus causes pronounced hyperglycemia (mouse age: 8 weeks; *n* = 6 per group). The Western blot to the right shows that G12D is present in liver lysates from WT mice injected with the G12D virus but not in liver lysates from WT mice treated with the eGFP virus. The HA-tagged G12D receptor was detected with anti-HA antibody. **b** WT and *Gna12−/−Gna13$^{fl/fl}$* mice were injected with the indicated AAV combinations (mouse age: 10 weeks; *n* = 6 per

group). Note that the hyperglycemic effects caused by CNO treatment of G12D-expressing mice (WT mice treated with AAVs coding for eGFP and Cre) is absent in G12D-expressing mice lacking Gα$_{12}$ and Gα$_{13}$ in their hepatocytes (*Gna12−/−Gna13$^{fl/fl}$* mice injected with AAVs coding for G12D and Cre). **c** Immunoblot showing the lack of Gα$_{13}$ expression in primary hepatocytes prepared from *Gna12−/−Gna13$^{fl/fl}$* mice treated with the AAV-TBG-Cre virus. Similar results were obtained in three additional independent experiments. All studies were carried out with male mice. Data represent means ± s.e.m. Numbers above horizontal bars refer to p values (panel a: two-tailed unpaired Student's t-test; (**b**): 2-way ANOVA followed by Bonferroni's post-hoc test). ns, no statistically significant difference. Source data are provided as a Source Data file.

require the activation of both hepatic G$_{12}$ and G$_{13}$ and that no other classes of heterotrimeric G proteins are involved in this effect.

## Both G$_{12}$ and G$_{13}$ are required for G12D-mediated hepatic glycogenolysis and gluconeogenesis

We next explored whether G12D-mediaed increases in the rates of glycogen breakdown and gluconeogenesis involved either G$_{12}$ or G$_{13}$ signaling, or both G protein subtypes. To address this issue, we initially

studied hepatic glycogenolysis using five different mouse strains (WT mice, Hep-G12D mice, Hep-G12D mice deficient in Gα$_{12}$ (whole body), Hep-G12D mice lacking Gα$_{13}$ in hepatocytes only, and Hep-G12D mice deficient in Gα$_{12}$ (whole body) and lacking Gα$_{13}$ in hepatocytes only) (Supplementary Fig. 9). Livers were harvested 30 min after CNO treatment (3 mg/kg, i.p.). We then used liver lysates to measure the activities of hepatic glycogen phosphorylase and Gsk3β, two enzymes that play central roles in promoting glycogenolysis. As expected, CNO

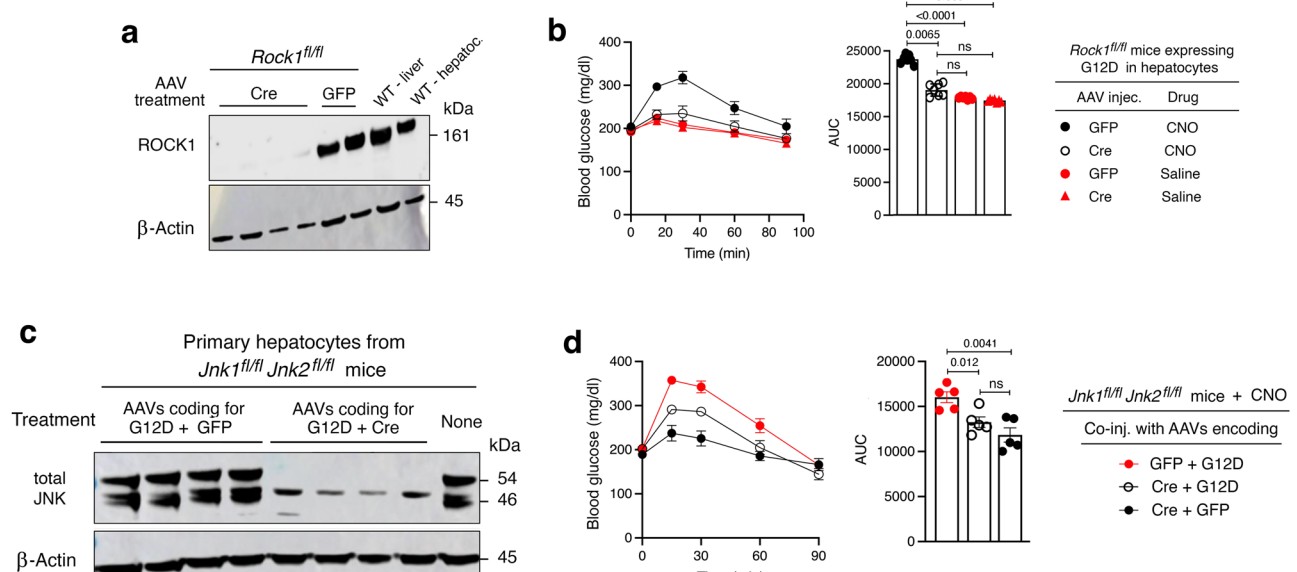

**Fig. 4 | ROCK1 and JNK signaling are required for G12D-mediated hyperglycemia.** *Rock1<sup>fl/fl</sup>*, *Jnk1<sup>fl/fl</sup> Jnk2<sup>fl/fl</sup>*, and all other mice used were maintained on regular chow for 8 weeks. Mice were then injected with either AAV-TBG-eGFP plus AAV-TBG-G12D or AAV-TBG-Cre plus AAV-TBG-G12D. One week later, freely fed mice were injected i.p. with CNO (3 mg/kg) or saline, followed by the measurement of blood glucose levels. **a** Immunoblot showing the lack of ROCK1 expression in primary hepatocytes prepared from *Rock1<sup>fl/fl</sup>* mice treated with the AAV-TBG-Cre virus. **b** The hyperglycemic effect caused by CNO treatment of Hep-G12D mice is absent in Hep-G12D mice lacking ROCK1 in hepatocytes. **c** Immunoblot showing the relative lack of JNK1/2 expression in primary hepatocytes prepared from *Jnk1<sup>fl/fl</sup> Jnk2<sup>fl/fl</sup>* mice treated with the AAV-TBG-Cre virus. **d** Inactivation of the *Jnk1* and *Jnk2* genes in hepatocytes leads to a marked reduction in the magnitude of CNO-induced hyperglycemic responses in Hep-G12D mice. Data are given as means ± s.e.m. (n = 6 mice/group). Numbers above horizontal bars refer to p-values (2-way ANOVA followed by Bonferroni's post-hoc test). ns, no statistically significant difference. Source data are provided as a Source Data file.

treatment of Hep-G12D mice caused a pronounced increase in the activities of both enzymes (Supplementary Fig. 9). These effects were strongly attenuated or abolished in CNO-treated Hep-G12D mice deficient in either $G\alpha_{12}$ or lacking $G\alpha_{13}$ in hepatocytes, indicating that both $G_{12}$ and $G_{13}$ signaling contribute to the G12D-mediated increase in hepatic glycogenolysis (Supplementary Fig. 9).

To explore the role of $G_{12}$ and $G_{13}$ signaling in stimulating hepatic gluconeogenesis, we prepared hepatocytes from the same five groups of mice described in the previous paragraph. We then incubated hepatocytes with CNO (10 µM) and high concentrations of two major gluconeogenic substrates (20 mM sodium lactate and 2 mM of sodium pyruvate, respectively) for 4 hr, followed by the measurement of glucose release. Expectedly, CNO treatment of hepatocytes prepared from Hep-G12D mice strongly stimulated glucose secretion (Supplementary Fig. 10). This response was greatly reduced in hepatocytes obtained from Hep-G12D mice deficient in either $G\alpha_{12}$ or lacking $G\alpha_{13}$ in hepatocytes (Supplementary Fig. 10). This observation indicates that both $G_{12}$ and $G_{13}$ signaling contribute to the G12D-mediated stimulation of hepatic gluconeogenesis.

### Hepatic ROCK1 activity is required for $G_{12/13}$-induced hyperglycemic responses

Rho-associated kinases (ROCKs) are members of the AGC family of serine/threonine kinases[33]. Two ROCK isoforms, ROCK1 and ROCK2, exist and they differ in their expression patterns, subcellular localization, and various functional properties[33]. Both ROCK proteins are major downstream effectors of the $G_{12/13}$/RhoA signaling pathway[34]. Previous studies have shown that ROCK1 is the predominant isoform expressed in the mouse liver[33,35] and that hepatic ROCK1 regulates the activity of several hepatic signaling proteins involved in glucose metabolism[36]. Moreover, Okin and Medzhitov[37] demonstrated that activation of ROCK signaling leads to a significant increase in HGP. Based on these previous studies, we explored the possibility that ROCK1 links increased $G_{12/13}$ activity to enhanced HGP. To address this

question, we initially generated Hep-G12D mice that lacked ROCK1 selectively in hepatocytes (strain name: Hep-G12D ROCK1 KO; see Methods for details) (Fig. 4a). We then injected Hep-G12D and Hep-G12D ROCK1 KO mice with CNO (3 mg/kg, i.p.) and monitored the resulting changes in blood glucose levels. Strikingly, the robust hyperglycemia observed with Hep-G12D mice was completely abolished in Hep-G12D ROCK1 KO mice (Fig. 4b). This observation clearly indicates that $G_{12/13}$-mediated ROCK1 activation is required for $G_{12/13}$-mediated increases in HGP.

### Inactivation of hepatic JNK signaling strongly impairs $G_{12/13}$-mediated hyperglycemia

Receptor-mediated stimulation of $G_{12/13}$ signaling also leads to the activation of JNK, resulting in various effects on gene expression and cellular functions[15–18]. In addition, previous work has shown that ROCK activation can promote JNK signaling[17,38,39]. We recently demonstrated that activation of hepatic JNK signaling strongly stimulates HGP[26]. Prompted by these observations, we speculated that $G_{12/13}$-dependent increases in HGP require ROCK1-depedendent activation of JNK signaling. To test this hypothesis, we used *Jnk1 fl/fl Jnk2 fl/fl* mice[40] to generate Hep-G12D mice that lacked both JNK1 and JNK2 selectively in hepatocytes (strain name: Hep-G12D JNK1/2 KO; see Methods for details) (Fig. 4c). As expected, CNO (3 mg/kg, i.p.) treatment of Hep-G12D mice resulted in pronounced increases in blood glucose levels (Fig. 4d). Interestingly, the magnitude of this response was significantly reduced, but not completely abolished, in Hep-G12D JNK1/2 KO mice (Fig. 4d), indicating that enhanced JNK signaling makes a major contribution to the hyperglycemic response caused by activation of hepatic $G_{12/13}$ signaling.

Western blotting studies with hepatocytes prepared from Hep-G12D JNK1/2 KO mice indicated that Cre-mediated JNK1/2 deletion was not 100% complete (Fig. 4c). This observation provides a possible explanation for the finding that CNO-treated Hep-G12D JNK1/2 KO mice still displayed a moderate increase in blood glucose levels (Fig. 4d).

## Role of JNK in rapid and prolonged G12D-mediated hepatic gluconeogenesis

Activation of JNK promotes the dephosphorylation of pFoxO1, resulting in the accumulation of the non-phosphorylated form of FoxO1 in the nucleus[41–43]. Nuclear FoxO1 acts as a transcription factor that strongly promotes the expression of gluconeogenic genes including *G6pc* and *Pepck*[2]. To investigate whether activation of G12D promotes the dephosphorylation of pFoxO1 in hepatocytes, we injected Hep-G12D mice and control littermates with CNO (3 mg/kg, i.p.). Thirty min later, mice were euthanized, and liver lysates were prepared and subjected to Western blotting studies. We found that CNO treatment of Hep-G12D mice caused a marked reduction in hepatic pFoxO1 levels, as compared to CNO-treated control littermates (Supplementary Fig. 11). This observation is consistent with a model in which G12D-mediated signaling promotes the dephosphorylation of hepatic pFoxO1, resulting in a prolonged increase in hepatic gluconeogenesis.

Under the same experimental conditions, the expression of pAkt, a key signaling hub in the insulin signaling pathway, was only slightly elevated in Hep-G12D mice (Supplementary Fig. 11), most likely due to enhanced insulin secretion following CNO-induced hyperglycemia (see Fig. 1h).

As shown in Fig. 2f, CNO treatment of G12D mice stimulated gluconeogenesis and glycogen breakdown already after 5 min, indicative of the involvement of non-transcriptional processes. To explore the nature of such potential non-transcriptional mechanisms, we studied the phosphorylation status of several signaling proteins and enzymes known to regulate hepatic glucose fluxes. Specifically, we injected Hep-G12D mice i.v. with either saline or CNO (3 mg/kg) via the inferior vena cava. Five min later, mice were euthanized, and liver lysates were prepared and subjected to Western blotting studies. We found that CNO-mediated activation of G12D led to the rapid inhibitory phosphorylation of liver glycogen synthase at position S641 and the activating phosphorylation of liver glycogen phosphorylase (PYGL) at position S15, respectively (Supplementary Fig. 12a, b). These rapid phosphorylation events provide a molecular basis for the quick onset of G12D-mediated glycogen breakdown.

Interestingly, while pJNK was barely detectable in livers from saline-treated Hep-G12D mice, CNO treatment of Hep-G12D mice led to a very robust increase in hepatic pJNK formation (Supplementary Fig. 12a, b). CNO-mediated rapid stimulation of G12D had only a small or no significant effect on the phosphorylation status of Akt and FoxO1, respectively, two key components of the insulin receptor signaling cascade (Supplementary Fig. 12a, c). We also found that CNO treatment of Hep-G12D mice led to a significant increase in the phosphorylation of hepatic Irs1 at S307 (Supplementary Fig. 12d). The formation of pIrs1 (S307) has been shown to be mediated by activated JNK (pJNK) and interferes with insulin receptor signaling[44–46]. Taken together, our data support the concept that G12D-mediated activation of JNK plays a central role in the rapid changes in hepatic glucose fluxes observed with CNO-treated Hep-G12D mice.

## In vitro studies with liver homogenates prepared from CNO-treated Hep-G12D mice

To demonstrate that the striking in vivo metabolic effects displayed by CNO-treated G12D mice were linked to changes in the activities of hepatic enzymes regulating hepatic glucose fluxes, we carried out studies with liver homogenates prepared from CNO-injected Hep-G12D and control mice. We found that hepatic preparations from CNO-treated Hep-G12D mice showed a marked increase in the activity of hepatic glycogen phosphorylase (Fig. 5a), the enzyme that catalyzes the rate-limiting step in glycogenolysis[47]. Moreover, the activities of glycogen synthase kinase 3β (GSK3β), which inactivates glycogen synthase via phosphorylation, and ROCK were also markedly elevated in liver homogenates from mice expressing G12D in hepatocytes (Fig. 5b, c).

Given the observation that CNO-treated Hep-G12D mice displayed a strong stimulation of hepatic glycogenolysis (Fig. 2e), we carried out additional in vitro studies aimed at identifying the cellular pathway involved in G12D-mediated increases in hepatic GP activity. Studies with liver homogenates prepared from CNO-treated Hep-G12D mice showed that the G12D-mediated increases in GP activity could be completely blocked by selective inhibitors of RhoA (rhosin) and ROCK (Y-27632) (10 μM each; Fig. 5d), strongly supporting the model that $G_{12/13}$-mediated stimulation of hepatic GP activity requires activation of the RhoA/ROCK signaling cascade.

Moreover, qRT-PCR studies showed that the expression levels of the rate-liming enzymes involved in stimulating HGP (glucose 6-phosphatase [*G6p*], fructose bisphosphatase [*Fbp*], and phosphoenolpyruvate carboxykinase 1 [*Pck1*]) were markedly increased in liver tissue prepared from CNO-treated Hep-G12D mice (Fig. 5e). This finding is consistent with the in vivo data shown in Fig. 2f, further corroborating the concept that activation of G12D signaling strongly promotes hepatic gluconeogenesis.

## CNO treatment of primary Hep-G12D hepatocytes strongly stimulates glucose release

To confirm that the G12D-mediated hyperglycemic effects observed in vivo were indeed caused by altered signaling in hepatocytes, we measured glucose release using primary hepatocytes isolated from Hep-G12D mice (Hep-G12D hepatocytes). As expected, CNO (10 μM) treatment of Hep-G12D hepatocytes, but not of control hepatocytes, resulted in a robust increase in glucose output (Fig. 6a, b). The magnitude of this effect was similar to that observed after incubation with glucagon (100 nM) (Fig. 6a), a hormone that is highly efficacious in stimulating HGP[2,3,48].

We also transduced primary hepatocytes prepared from WT control mice (genetic background: C57BL/6) and *Gna12*$^{-/-}$ mice with the same genetic background with adenoviruses coding for eGFP (control virus) or G12D (Ad-G12D), respectively. One day after virus treatment, cells were incubated with CNO (10 μM) or glucagon (100 nM; positive control), and the amount of glucose released into the medium was determined after a 5 hr incubation period. As expected, CNO treatment of eGFP-expressing WT hepatocytes had no significant effect on glucose release (Fig. 6b). In contrast, and in agreement with the data shown in Fig. 6a, CNO treatment of G12D-expressing WT hepatocytes resulted in a pronounced increase in glucose output that was comparable in magnitude to the corresponding glucagon response (Fig. 6b). In contrast, CNO was unable to promote glucose release from G12D-expressing hepatocytes prepared from *Gna12*$^{-/-}$ mice (Fig. 6b), clearly indicating that $G_{12}$ signaling is required for G12D-induced stimulation of glucose release from hepatocytes.

## CNO treatment of Hep-G12D hepatocytes has no effect on intracellular cAMP levels

It is well known that glucagon and other agents that stimulate the production of cAMP in hepatocytes lead to pronounced increases in HGP[2,49]. Previous studies have shown that receptor-induced $G_{12/13}$ signaling can increase cytoplasmic cAMP levels via activation of adenylyl cyclase isoform 7 (AC7) in certain cell types[50,51]. To explore whether this pathway is operative in mouse hepatocytes, we treated primary hepatocytes prepared from Hep-G12D mice with CNO (10 μM), followed by the monitoring of intracellular cAMP levels. We found that CNO treatment of G12D-expressing hepatocytes had no significant effect on cytoplasmic cAMP levels (Supplementary Fig. 13). In contrast, forskolin (10 μM; positive control) induced a very robust cAMP response in these cells (Supplementary Fig. 13). These data indicate that changes in intracellular cAMP levels do not play a role in the rapid increase in HGP observed after simulation of hepatic $G_{12/13}$ signaling.

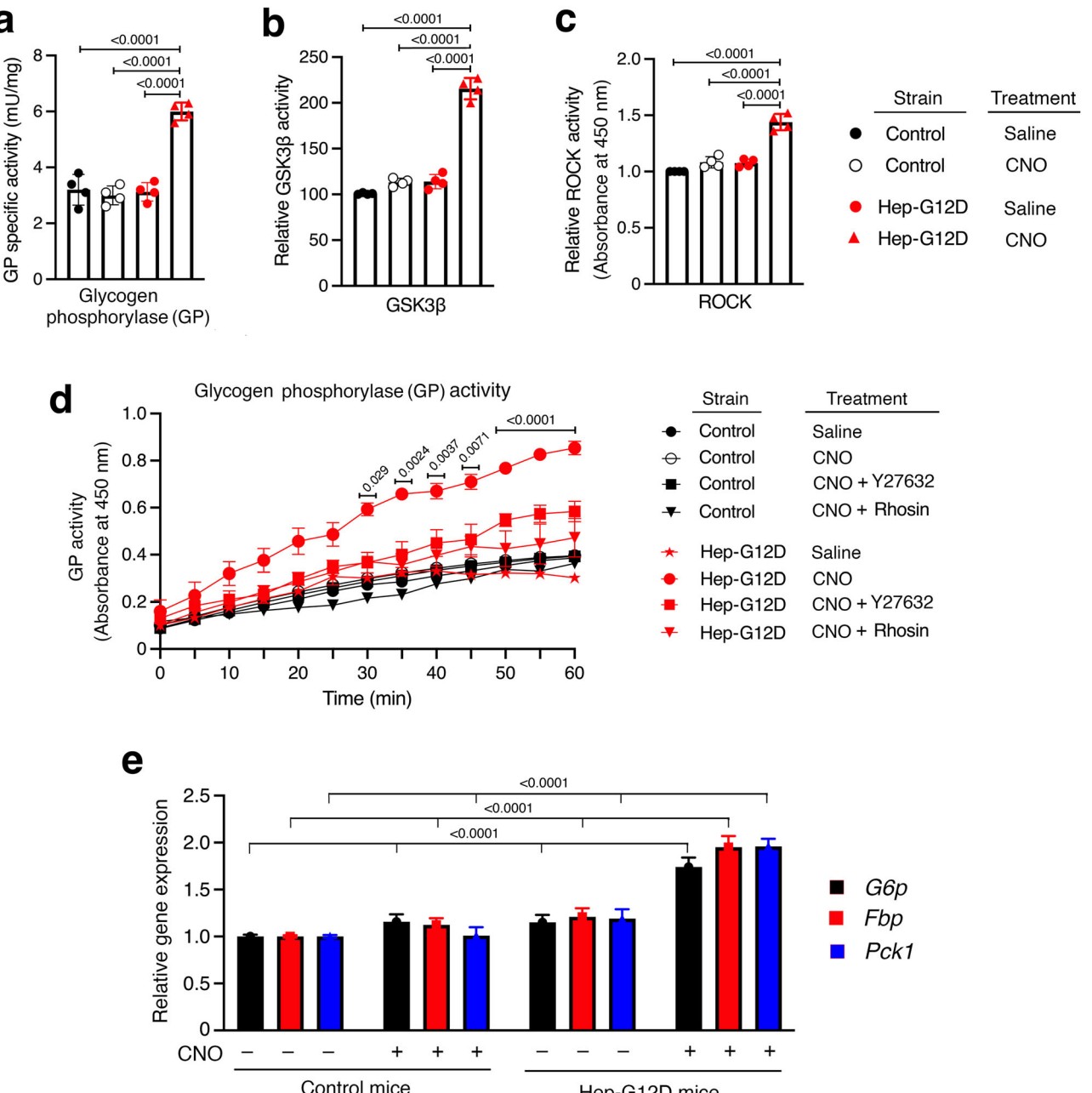

**Fig. 5 | Hepatic enzyme activity assays and gene expression analysis.** All experiments were carried out with 12-week-old male Hep-G12D mice and control littermates maintained on regular chow. Assays were performed with liver tissue homogenates obtained from mice 30 min after i.p. injection with CNO (3 mg/kg) or saline. (**a–c**), Enzyme activity assays. The activities of hepatic glycogen phosphorylase (GP) (**a**), GSK3β (**b**), and ROCK (**c**) were determined. **d** GP activity kinetic assay using mouse primary hepatocytes from Hep-G12D mice and control littermates. The following drugs were used: CNO (10 μM), Y27632 (ROCK inhibitor, 10 μM), and

rhosin (RhoA inhibitor, 10 μM). **e** Expression levels of key genes regulating hepatic glucose metabolism using liver RNA obtained from mice 30 min after i.p. injection with CNO (3 mg/kg) or saline (-). Gene expression levels were obtained via qRT-PCR and normalized relative to β-actin RNA expression. Data represent means ± s.e.m. ($n = 6$ mice/group). Numbers above horizontal bars refer to p values. Statistical significance was determined by 2-way ANOVA followed by Bonferroni's post-hoc test. Source data are provided as a Source Data file.

## G12D activation of the ROCK/JNK signaling cascade promotes glucose release from primary hepatocytes

Previous work demonstrated that activated $G_{12/13}$ signaling leads to the activation of ROCK[8,9]. In agreement with this finding, we showed that the CNO-induced stimulation of glucose release from primary hepatocytes prepared from Hep-G12D mice (Hep-G12D hepatocytes) was greatly reduced by a selective ROCK inhibitor (Y-27632, 10 μM) (Fig. 6a), consistent with the in vivo data shown in Fig. 4b. Moreover, incubation of Hep-G12D hepatocytes with a JNK inhibitor (SP600125, 10 μM) also strongly inhibited the stimulatory effect of CNO on

glucose release (Fig. 6a), in agreement with the outcome of vivo studies shown (Fig. 4d). Control experiments showed that treatment of Hep-G12D hepatocytes with Y-27632 or SP600125 alone (10 μM each) had no significant effect on glucose secretion (Supplementary Fig. 14).

We also demonstrated that CNO treatment of Hep-G12D hepatocytes resulted in a pronounced increase in JNK phosphorylation (Fig. 6c). This effect could be completely abolished by co-incubation with Y27632 (10 μM; Fig. 6c), further supporting the concept that ROCK-dependent JNK activation plays a central role in $G_{12/13}$-induced stimulation of HGP. Taken together, both the in vivo and in vitro data

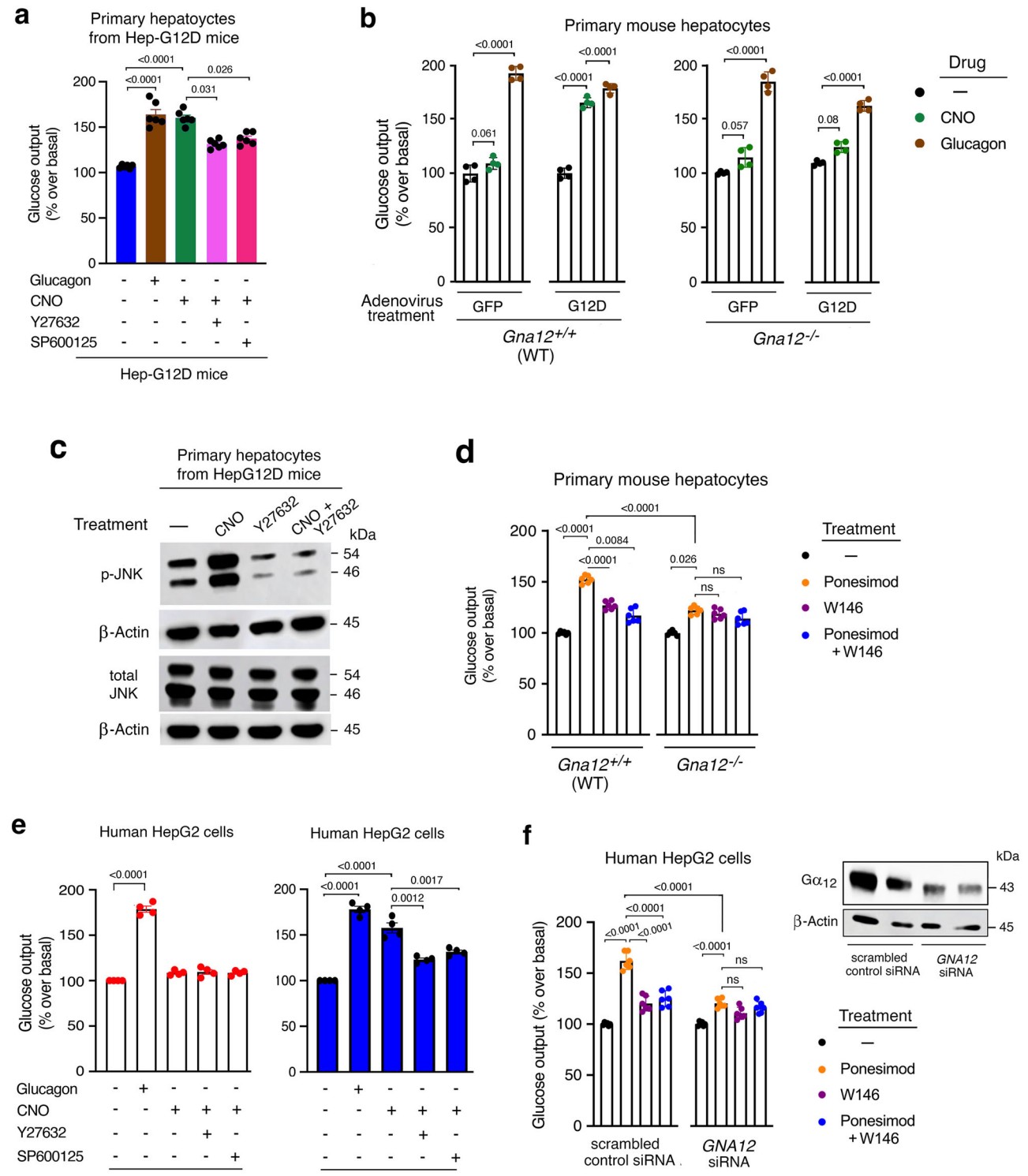

strongly support the existence of a $G_{12/13}$-ROCK-JNK signaling pathway that promotes HGP with high efficacy.

As mentioned earlier, it is well known that stimulation of ROCK1 can promote the phosphorylation and activation of JNK in different cell types[17,38,39]. Previous work has shown that ROCK1 does not phosphorylate JNK directly but that this phosphorylation event is mediated indirectly by other kinases acting downstream of ROCK1, whose molecular identity remains to be identified[17]. Thus, it is likely that a similar mechanism is operative in mouse hepatocytes.

### Activation of a $G_{12}$-coupled receptor endogenously expressed by mouse hepatocytes

Accumulating evidence suggests that sphingosine 1-phosphate (S1P), a bioactive sphingolipid derivative, plays an important role during the pathogenesis of obesity and T2D[52]. Previous studies have shown that the activation of hepatocyte S1P receptors has multiple effects on hepatocyte function[52]. Interestingly, recent studies demonstrated that the $S1P_1$ receptor subtype (S1PR1), which is highly expressed in the liver[4], can couple to G proteins of the $G_{12/13}$ family with high efficacy[11]. To explore the effect of activating S1PR1s endogenously expressed by

**Fig. 6 | Glucose output assays carried out with primary mouse hepatocytes and human HepG2 cells. a** Primary hepatocytes prepared from Hep-G12D mice were incubated in the absence or presence of CNO (10 μM), and glucose outflow was examined 5 h later. Cells were also treated with glucagon (100 nM), CNO plus Y-27632 (ROCK inhibitor), and CNO plus SP600125 (JNK inhibitor) (10 μM each drug) ((n = 6 mice per group). **b** Primary hepatocytes prepared from WT or *Gna12*−/− mice injected with the indicated adenoviruses were incubated with CNO (10 μM) or glucagon (100 nM), followed by the measurement of total glucose secretion for 5 h later (n = 4 independent experiments). **c** Western blot analysis of G12D-mediated JNK activation. Primary hepatocytes obtained from Hep-G12D mice were treated with CNO and other drugs. A representative blot is shown. Two additional independent experiments yielded similar results. **d** Ponesimod-induced glucose production is abolished in mouse hepatocytes with Gα$_{12}$ deficiency. Primary hepatocytes from 9 to 10-week old WT (C57BL/6) and *Gna12*−/− mice were incubated in the presence of ponesimod (10 μM; S1PR1 agonist), either alone or in the presence of W146 (1 μM), a selective S1PR1 antagonist. Total glucose output was measured 5 h later (n = 6 mice per group). **e** Glucose output assays with human HepG2 cells. HepG2 cells infected with the indicated adenoviruses were incubated with CNO and other drugs. Glucose secretion was determined after a 3 h incubation period. Data are from four independent experiments, each carried out in triplicate. **f** Ponesimod-induced glucose output is nearly abolished in HepG2 cells following treatment with *GNA12* siRNA. HepG2 cells were treated with either scrambled control siRNA or *GNA12* siRNA. Cells were then incubated in the presence of ponesimod (10 μM), either alone or in the presence of W146 (1 μM), followed by glucose output measurements 3 h later. Data are from three independent experiments, each carried out in triplicate. The insert shows a representative Western blot examining the expression of Gα$_{13}$. Data are given as means ± s.e.m. Numbers above horizontal bars refer to p values (2-way ANOVA followed by Bonferroni's post-hoc test). ns, no statistically significant difference. Source data are provided as a Source Data file.

hepatocytes on glucose secretion, we acutely treated hepatocytes isolated from WT mice with ponesimod, a selective S1PR1 agonist[53]. As shown in Fig. 6d, ponesimod (10 μM) treatment resulted in a significant increase in glucose release from WT hepatocytes. This effect was nearly abolished by a selective S1PR1inhibitor (W146; 1 μM)[54] in WT hepatocytes and greatly reduced in Gα$_{12}$-deficient hepatocytes (primary hepatocytes prepared from *Gna12*−/− mice), indicating that ponesimod promotes HGP via activation of G$_{12}$-coupled S1PR1s (Fig. 6d). This observation is consistent with the recent finding that activation of S1PR1s can recruit G$_{12}$, but not G$_{13}$, with high efficacy[11].

In vivo studies showed that ponesimod treatment (20 mg/kg, i.p.) of WT mice resulted in a mild but significant increase in blood glucose levels, as compared to saline-treated WT mice (Supplementary Fig. 15a). This effect was absent in mice deficient in either Gα$_{12}$ (whole body) or both Gα$_{12}$ (whole body) and Gα$_{13}$ (hepatocytes only) (Supplementary Fig. 15b, c). Thus, the outcome of these in vivo studies is consistent with the in vitro data obtained with isolated hepatocytes (Fig. 6d).

### In vitro studies with a human hepatocyte cell line (HepG2)
We next examined whether activation of the G$_{12/13}$-ROCK-JNK signaling cascade caused a similar increase in glucose production in human hepatocytes. For these studies, we used human liver carcinoma-derived HepG2 cells[55]. We infected HepG2 cells with an adenovirus coding for G12D (G12D-HepG2 cells) or with a control adenovirus coding for eGFP (GFP-HepG2 cells), respectively. We then treated these cells with either vehicle or CNO (10 μM) and measured glucose release into the medium during a 3 h incubation period. As shown in Fig. 6e, CNO treatment of G12D-HepG2 cells, but not of GFP-HepG2 cells, resulted in a significant increase in glucose output. As observed with mouse hepatocytes, this response was almost completely abolished after treatment of G12D-HepG2 cells with pharmacological inhibitors of ROCK (Y-27632) or JNK (SP600125) (10 μM each; Fig. 6e).

To examine whether activation of S1PR1s endogenously expressed by HepG2 cells resulted in similar effects on glucose secretion, we initially incubated HepG2 cells that had been exposed to scrambled control siRNA with ponesimod (10 μM) for 3 hr. Under these conditions, ponesimod triggered a significant increase in glucose release from HepG2 cells (Fig. 6g). This effect was greatly reduced in the presence of a selective S1PR1 inhibitor (W146; 1 μM), indicative of the involvement of S1PR1s (Fig. 6f).

To confirm the involvement of G$_{12}$ signaling in this ponesimod response, we used *GNA12* siRNA to knock down the expression of Gα$_{12}$ (Fig. 6f). We found that the stimulatory effect of ponesimod on glucose release in control HepG2 cells was almost completely abolished after treatment of cells with *GNA12* siRNA (Fig. 6f). These data indicate that the G$_{12/13}$-ROCK-JNK signaling module promotes glucose secretion in both mouse and human hepatocytes.

### Fasting increases hepatic *Gna12* and *Rock1* expression levels and ROCK1 activity in a G$_{12/13}$-depedent fashion
Since gluconeogenesis is elevated under fasting conditions[2,3], we next investigated whether hepatic *Gna12* and/or *Gna13* expression levels were altered in mice after a 24 h fast. We found that transcript levels of *Gna12*, but not of *Gna13*, were significantly increased in fasted WT mice (~2-fold; Fig. 7a) (also see ref. 20). Moreover, hepatic *Rock1* mRNA levels were increased by 2-3-fold under these conditions. (Fig. 7a). *RhoA* expression levels were also elevated in food-deprived mice, but this effect failed to reach statistical significance (Fig. 7a). Moreover, under these experimental conditions, we observed a pronounced increase in hepatic *Rock1* mRNA levels (Fig. 7b), which was associated with a marked increase in hepatic ROCK activity (Fig. 7c).

Strikingly, the fasting-induced increases in hepatic *Rock1* expression and ROCK activity were absent in mice lacking Gα$_{12}$ (whole body) and Gα$_{13}$ in hepatocytes (*Gna12*−/− *Gna13* fl/fl mice treated with AAV-TGB-Cre; short name: Hep-G12/G13 KO mice) (Fig. 7b–d). This finding strongly supports the concept that enhanced G$_{12/13}$-ROCK1 signaling contributes to the maintenance of euglycemia under fasting conditions.

### Metabolic studies with mice lacking G$_{12/13}$ signaling in hepatocytes
We next investigated whether glucose homeostasis, including fasting blood glucose levels and glucose tolerance, was altered in Hep-G12/G13 KO mice. When maintained on regular chow, Hep-G12/G13 KO mice showed similar fasting blood glucose levels (Supplementary Fig. 16a, b; time points '0') and glucose tolerance, as compared to their control littermates (age-matched WT mice or *Gna12*−/− *Gna13* fl/fl mice treated with AAV-TGB-eGFP, respectively) (Supplementary Fig. 16a, b).

Likewise, Hep-G12/G13 KO mice maintained on a HFD for 8 weeks did not differ from their control littermates in fed and fasting blood glucose and plasma insulin levels (Supplementary Fig. 16c). We also analyzed mice that had been maintained on a HFD for a longer period of time (16 weeks). Under these experimental conditions, G12/G13 KO mice and their control littermates showed similar glucose and insulin tolerance (Supplementary Fig. 16d, e). Strikingly, however, Hep-G12/G13 KO mice showed significantly reduced fasting blood glucose levels upon prolonged HFD feeding (Supplementary Fig. 16f).

We next considered the possibility that compensatory hepatic signaling via other classes of heterotrimeric G proteins may partially mask the metabolic phenotypes displayed by Hep-G12/G13 KO mice. Since activation of hepatocyte G$_s$ and G$_{q/11}$ signaling strongly promotes HGP[56,57], we investigated whether the hepatic expression levels of Gα$_s$ and Gα$_{q/11}$ were altered in Hep-G12/G13 KO mice. Western blotting studies showed that hepatic Gα$_s$ and Gα$_{q/11}$ protein levels were similar in Hep-G12/G13 KO and control mice maintained on the HFD for 16 weeks (Supplementary Fig. 16g), indicating that the lack of hepatic

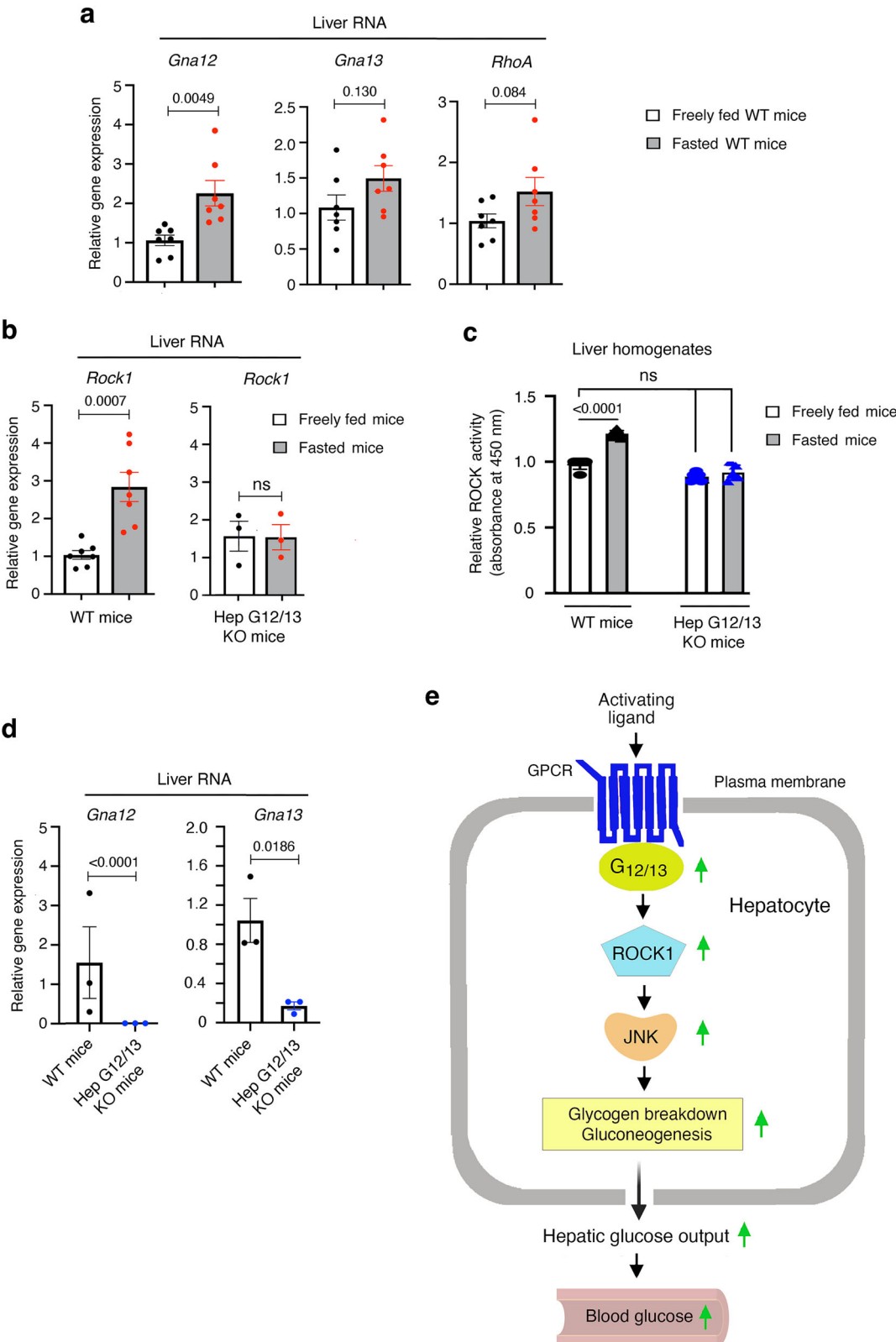

**Fig. 7 | Fasting increases hepatic *Gna12* and *Rock1* expression levels and ROCK1 activity in WT mice.** Total mRNA was isolated from 10-week-old WT mice (males) that had free access to food or that had been fasted for 24 h. **a, b** The expression of *Gna12* and *Rock1* are upregulated in fasted mice, as determined via qRT-PCR. **c** Fasted WT mice also show a significant increase in hepatic ROCK activity, as studied with liver homogenates. **b**–**d** The fasting-induced increases in hepatic *Rock1* expression and ROCK activity are abolished in Hep-G12/13 KO mice (whole body

$G\alpha_{12}$ KO mice lacking $G\alpha_{13}$ selectively in hepatocytes; see Methods for details). **e** Scheme depicting how receptor-mediated activation of hepatic $G_{12/13}$ signaling promotes glucose output from hepatocytes. Data are given as means ± s.e.m. ($n = 7$/group). Numbers above horizontal bars refer to *p*-values (two-tailed unpaired Student's t-test (**a, b, d**); 2-way ANOVA followed by Bonferroni's post-hoc test (**c**)). Source data are provided as a Source Data file.

$G_{12/13}$ signaling does not affect the hepatic expression levels of $G\alpha_s$ and $G\alpha_{q/11}$.

### *GNA12* expression levels in human liver samples correlate with insulin resistance

Finally, we assessed hepatic gene expression levels of *GNA12* and *GNA13* using liver biopsy samples obtained from human subjects with metabolic dysfunction-associated steatotic liver disease (MASLD; $n = 24$). Interestingly, hepatic *GNA12* expression levels, as determined via RNAseq, were positively correlated with HOMA-IR (homeostatic model assessment for insulin resistance; $r = 0.41$, $p = 0.05$) and negatively correlated with Si (insulin sensitivity index obtained from frequently-sampled intravenous glucose tolerance test [FS IVGTT]; $r = -0.42$, $p = 0.05$) (Supplementary Fig. 17). In contrast, hepatic *GNA13* expression levels showed no correlation with any of the metabolic parameters analyzed (Supplementary Fig. 17).

## Discussion

The liver plays a central role in regulating blood glucose levels[47,58]. An increase in HGP is a major contributor to the elevated blood glucose levels characteristic of T2D and related metabolic disorders[47]. HGP is regulated, to a major extent, by the activity of various hormones and neurotransmitter that act on GPCRs present on the cell surface of hepatocytes[2,3,5]. The ability of $G_s$-, $G_i$-, and $G_q$-type G proteins to modulate hepatic glucose fluxes has been explored in considerable detail in the past[5,26,56,57]. In contrast, little is known about whether receptor-mediated activation of hepatocyte $G_{12/13}$ signaling affects key pathways that regulate HGP.

To address this question, we initially used a chemogenetic strategy involving the selective expression of a $G_{12/13}$-coupled designer GPCR (G12D) in hepatocytes of mice (Hep-G12D mice). Strikingly, treatment of Hep-G12D mice with CNO, a small molecule that can selectively activate G12D and related designer receptors of the DREADD family[23,24] resulted in pronounced increases in blood glucose levels (Fig. 1). This effect was absent in mice lacking $G\alpha_{12}$ and $G\alpha_{13}$ in hepatocytes (*Gna12−/− Gna13 fl/fl* mice treated with the AAV-TBG-Cre virus) (Fig. 3b), indicating that activation of hepatocyte $G_{12/13}$ signaling strongly promotes HGP. Hepatic glucose flux studies with conscious mice demonstrated that the $G_{12/13}$-medatiated hyperglycemia involves increases in the rates of both hepatic gluconeogenesis and glycogen breakdown (Fig. 2). In vivo and in vitro studies with different mutant mouse models showed that both $G_{12}$ and $G_{13}$ signaling contribute to the G12D-mediated increases in hepatic gluconeogenesis and glycogenolysis (see, for example, Supplementary Figs. 9 and 10).

Metabolic studies with additional mutant mouse strains indicated that $G_{12/13}$-mediated increases in blood glucose levels required the activation of ROCK1 and JNK (Fig. 4). Consistent with this observation, ROCK1/2 represents a major effector protein activated following receptor-mediated stimulation of $G_{12/13}$ signaling[34]. Moreover, several studies have shown that activated ROCK1/2 can stimulate the phosphorylation and activation of JNK (reviewed in ref. 38).

We recently demonstrated that activation of G proteins of the $G_i$-family expressed by hepatocytes also enhances the activity of hepatic JNK, resulting in elevated HGP and hyperglycemia[26]. Stimulation of hepatic JNK signaling leads to the increased expression of genes that play key roles in promoting HGP[26,59,60]. In agreement with this observation, we demonstrated that $G_{12/13}$-dependent activation of hepatocyte JNK signaling was associated with increases in the expression levels of *G6p*, *Fbp*, and *Pck1*, the rate-limiting enzymes in promoting HGP (Fig. 5e). Consistent with this finding, stimulation of $G_{12/13}$ signaling resulted in a marked increase in glucose release in a ROCK- and JNK-dependent fashion in primary mouse hepatocytes and human HepG2 cells (Fig. 6).

CNO treatment of Hep-G12D mice stimulated gluconeogenesis and glycogenolysis already after 5 min after CNO administration

(Fig. 2e, f), indicative of non-transcriptional mechanisms involved in these processes. Activation of hepatic $G_{12/13}$ signaling caused pronounced increases in the activities of hepatic glycogen phosphorylase and Gsk3β (Fig. 5). In addition, Western blotting studies showed that CNO-mediated activation of G12D led to the rapid (within 5 min) inhibitory phosphorylation of liver glycogen synthase at position S641 and the activating phosphorylation of liver glycogen phosphorylase at position S15, respectively (Supplementary Fig. 12a, b). These observations provide a cellular basis for the quick onset of G12D-mediated glycogen breakdown.

As discussed in the past, rapid hepatic gluconeogenesis (within minutes) can be caused by many transcription-independent mechanisms[2,61–63], including the allosteric inhibition of key regulatory glycolytic enzymes, the activation of pathways that alter the availability of gluconeogenetic substrates (lactate, amino acids, glycerol, etc.), the cytoplasmic redox state, or the activity of various phosphodiesterases and other enzymes or signaling pathways that regulate signaling via hepatic glucagon and insulin receptors[2,61,62]. Additional factors that could contribute to transcription-independent increases in gluconeogenesis include changes in posttranslational modifications of existing signaling complexes or altered function of different components of the mitochondrial electron transport chain[2,61,62]. We observed that CNO treatment of Hep-G12D mice led to a very rapid (with 5 min) and pronounced increase in hepatic pJNK formation (Supplementary Fig. 12a, b). Since pJNK can act on more than 100 different cellular substrates including various protein phosphatases and kinases[64], it is likely that the rapid G12D-mediated formation of pJNK makes a major contribution to the quick increase in hepatic glucose output observed with CNO-treated Hep-G12D mice (Fig. 2f). CNO treatment of Hep-G12D mice also led to a significant increase in the phosphorylation of Irs1 at Ser307 (Supplementary Fig. 12d). Previous studies have shown that this phosphorylation step is mediated by activated JNK (pJNK)[44–46], resulting in impaired insulin receptor signaling. Taken together, our data support the concept that G12D-mediated activation of JNK plays a central role in the rapid stimulation of gluconeogenesis observed with CNO-treated Hep-G12D mice. We are planning to systematically explore the cellular substrates and signaling mechanisms through which $G_{12/13}$-dependent hepatic JNK activation affects hepatic glucose metabolism in a detailed follow-up study.

In contrast to our findings, Kim et al.[21] reported that hepatocyte-specific $G\alpha_{13}$ KO mice showed hyperglycemia and impaired glucose tolerance under different experimental conditions. Whereas we used a virus-based approach that led to the inactivation of the *Gna13* gene in hepatocytes of adult mice, the $G\alpha_{13}$ mutant mice generated by Kim et al.[21] involved crossing *Albumin-Cre* transgenic mice to *Gna13 fl/fl mice*. Since the albumin promoter is active during early development[65], and $G\alpha_{13}$ plays a key role in important developmental processes[66–68], it is possible that the outcome of the study by Kim et al.[21] was affected by compensatory developmental changes. Kim et al.[21] also reported that hepatic $G\alpha_{13}$ expression levels determined with human liver biopsy samples were inversely correlated with indices of diabetes including insulin resistance determined via HOMA-IR. In contrast, we failed to detect a significant relationship between hepatic *GNA13* levels and HOMA-IR, glucose tolerance, and related metabolic parameters in human liver samples (Supplementary Fig. 17). One possible explanation for this discrepant finding is that the patient cohorts analyzed in the two studies differed in genetic/ethnic background (the study by Kim et al. did not indicate the ethnicity of the subjects from which the liver biopsies were taken). In contrast, we found that hepatic *GNA12* expression levels were positively correlated with HOMA-IR and negatively correlated with FS IVGTT Si (Supplementary Fig. 17c, e), consistent with the concept that elevated hepatic *GNA12* expression levels contribute to impaired insulin sensitivity and glucose homeostasis in humans.

Several of our findings clearly indicate that hepatic $G_{12/13}$ signaling is of physiological relevance. For example, we found that the fasting-induced increases in hepatic *Rock1* expression and ROCK activity required hepatic $G_{12/13}$ signaling (Fig. 7b–d). We also found that fasted Hep-G12/G13 KO mice consuming a HFD for an extended period of time (16 weeks) showed significantly lower blood glucose levels than their control littermates (Supplementary Fig. 16f), strongly suggesting that hepatic $G_{12/13}$ signaling plays a role in maintaining euglycemia under certain nutritional conditions. Moreover, we carried out in vitro and in vivo studies targeting endogenously expressed hepatic $S1P_1$ receptors (S1PR1s) which can interact with G proteins of the $G_{12/13}$ family with high efficacy[11]. In vitro studies showed that activation of hepatocyte S1PR1s stimulated glucose release in a $G_{12}$-dependent fashion (Fig. 6d, f). In agreement with this observation, in vivo studies demonstrated that treatment of WT mice with ponesimod, a selective S1PR1 agonist, increased blood glucose levels in WT mice but not in Hep-G12/13 KO mice (Supplementary Fig. 15). Finally, studies with human liver samples revealed that hepatic *GNA12* (encoding $G\alpha_{12}$) expression levels positively correlated with parameters of insulin resistance and impaired glucose homeostasis (Supplementary Fig. 17), in agreement with a potential pathophysiological role of increased hepatic $G_{12/13}$ signaling. Taken together, a strong body of data indicates that hepatic $G_{12/13}$ signaling plays a physiological role in regulating hepatic glucose fluxes and glucose homeostasis.

In conclusion, metabolic studies with $G_{12/13}$ mutant mice support the concept that activation of hepatocyte $G_{12/13}$ signaling promotes HGP, resulting in pronounced changes in whole body glucose homeostasis. The pathway linking enhanced $G_{12/13}$ signaling to increased HGP involves stimulation of the ROCK1/JNK signaling cascade in both mouse and human hepatocytes (Fig. 7c, e). Given the fact that the liver expresses dozens of $G_{12/13}$-coupled receptors[4,11], our study may lead to the development of drugs that target one or more of these receptors for the treatment of diseases (e.g. T2D) characterized by pathologically elevated HGP.

## Methods

### Study approval
All animal studies were approved by the NIDDK Institutional Animal Care and Use Committee (NIH, Bethesda, MD). Studies involving human subjects were approved by the NIDDK/NIAMS Institutional Review Board, and all subjects provided written informed consent.

### Drugs, reagents, commercial kits, and antibodies
The sources of all drugs, reagents, commercial kits, antibodies, and mouse strains are listed in Supplementary Table 1.

### Mouse maintenance and diet
All mice were housed in a pathogen-free barrier facility at 23 °C with a 12 h light/12 h dark cycle (light period: 6:00 am to 6:00 pm). Mice consumed either regular mouse chow (7022 NIH-07, 15% kcal fat, energy density 3.1 kcal/g, Envigo Inc.) or a high-fat diet (HFD; F3282, 60% kcal fat, energy density 5.5 kcal/g, Bioserv) with unlimited access to water.

### Recombinant viruses
Adeno-associated viruses (AAVs; serotype 8) coding for Cre recombinase (AAV-TBG-Cre) or eGFP (AAV-TBG-eGFP) were obtained from the Vector Core of the University of Pennsylvania (Philadelphia, PA, USA) or Addgene (Watertown, MA, USA). AAV-TBG-Cre directs the selective expression of Cre recombinase in hepatocytes (Cre expression is under the transcriptional control of the hepatocyte-selective thyroxine-binding globulin (TBG) promoter). The AAV-TBG-eGFP virus, which codes for eGFP, was used for control purposes. We also generated an AAV construct in which the G12D coding sequence was inserted into the AAV-TBG plasmid, yielding AAV-TBG-G12D. Viral particles

(serotype 8) were generated by VectorBuilder Inc. (Chicago, IL, USA). Adenoviruses coding for HA-G12D (Ad-CMV-HA-G12D) or eGFP (Ad-CMV-eGFP) were custom-produced by Vector Biolabs (Malvern, PA, USA).

### Generation of mutant mouse strains
All mutant mice used in this study were maintained on a C57BL/6 genetic background. The generation of *Rosa26-LSL-G12D-IRES-GFP* mice has been reported recently[22]. For the sake of simplicity, we refer to these mice as LSL-G12D mice throughout the text. To remove the LSL cassette and to induce the hepatocyte-selective expression of G12D, LSL-G12D mice (males or females; age: 8 weeks) were injected into the tail vein with the AAV-TBG-Cre virus ($1.5 \times 10^{11}$ viral particles per mouse suspended in 100 µl of saline). For the sake of brevity, we refer to the AAV-TBG-Cre-treated LSL-G12D mice as Hep-G12D mice throughout the manuscript. To generate control littermates, LSL-G12D mice were injected in the same fashion with the AAV-TBG-eGFP control virus.

An alternative strategy to generate Hep-G12D mice involved the treatment of WT mice with an AAV coding for G12D (AAV-TBG-G12D). In this case, 8-week-old male WT mice (C57BL/6 mice, Taconic) were injected into the tail vein with AAV-TBG-G12D ($1 \times 10^{11}$ viral particles/mouse suspended in 100 µl saline). Control littermates (WT mice without hepatic G12D expression) were generated by using the same strategy but by replacing the AAV-TBG-G12D virus with AAV-TBG-eGFP which is pharmacologically inert.

We also injected $Gna12^{-/-}Gna13^{fl/fl}$ mice[69] (8-week-old males) with either AAV-TBG-Cre or AAV-TBG-eGFP. The mice treated with the AAV-TBG-Cre virus lacked $G\alpha_{13}$ selectively in hepatocytes and $G\alpha_{12}$ throughout the body (abbreviated strain name: Hep-G12/13 KO). $Gna12^{-/-}Gna13^{fl/fl}$ mice treated with AAV-TBG-eGFP lacked $G\alpha_{12}$ throughout the body but showed normal $G\alpha_{13}$ expression. We also co-injected $Gna12^{-/-}Gna13^{fl/fl}$ mice with either AAV-TBG-Cre plus AAV-TBG-G12D or AAV-TBG-eGFP plus AAV-TBG-G12D. Co-injection with AAV-TBG-Cre plus AAV-TBG-G12D yielded mice that expressed the G12D receptor selectively in hepatocytes but lacked $G\alpha_{12}$ throughout the body and did not express $G\alpha_{13}$ in hepatocytes (abbreviated strain name: Hep-G12D G12/13 KO). Co-injection of $Gna12^{-/-}Gna13^{fl/fl}$ mice with AAV-TBG-eGFP plus AAV-TBG-G12D resulted in mice that expressed the G12D receptor selectively in hepatocytes, lacked $G\alpha_{12}$ throughout the body, but showed normal expression of $G\alpha_{13}$ (abbreviated strain name: Hep-G12D G12 KO).

We also co-injected 8-week-old male $Rock1^{fl/fl}$ mice[70] with either AAV-TBG-Cre plus AAV-TBG-G12D or AAV-TBG-eGFP plus AAV-TBG-G12D, respectively ($1 \times 10^{11}$ viral particles per virus and mouse). Co-injection with AAV-TBG-Cre plus AAV-TBG-G12D yielded mice expressing the G12D receptor selectively in hepatocytes but did not express ROCK1 in hepatocytes. Co-treatment of $ROCK1^{fl/fl}$ mice with AAV-TBG-eGFP plus AAV-TBG-G12D resulted in mice expressing G12D selectively in hepatocytes that showed normal hepatic expression of ROCK1. For each virus, $1 \times 10^{11}$ viral particles per mouse were injected.

In addition, we generated Hep-G12D mice that lacked JNK1 and JNK2 selectively in hepatocytes (strain name: Hep-G12D JNK1/2 KO). To obtain this strain, we co-injected (via the tail vein) 8-week-old male $Jnk1^{fl/fl}Jnk2^{fl/fl}$ mice[40] with a mixture of AAV-TBG-G12D and AAV-TBG-Cre ($1 \times 10^{11}$ viral particles per virus and mouse). To obtain a proper cohort of control mice, the $Jnk1^{fl/fl}Jnk2^{fl/fl}$ mice were subjected to the same procedure, except that the Cre virus was replaced with AAV-TBG-eGFP ($1 \times 10^{11}$ viral particles per virus and mouse). These mice expressed G12D selectively in hepatocytes but showed normal hepatic expression levels of JNK1 and JNK2.

### In vivo metabolic tests
Mouse phenotyping studies were initiated two weeks after treatment of mice with AAVs. In vivo metabolic tests were performed with male or

female mice (age range: 10-20 weeks) using standard procedures. In brief, prior to i.p. glucose tolerance tests (IGTT), mice were fasted overnight for ~12 h. Blood glucose levels were determined using blood collected from the tail vein immediately before and at defined time points after i.p. injection of glucose (1 or 2 g/kg, as indicated). For pyruvate and insulin tolerance tests (PTT and ITT, respectively), mice were fasted for 12 and 4 h, respectively, and then injected i.p. with either sodium pyruvate (1 g/kg) or human insulin (0.75–1.5 U/kg; Humulin, Eli Lilly), respectively. Blood glucose levels were measured at defined post-injection time points.

To study glucose-stimulated insulin secretion (GSIS), Hep-G12D or control mice (10–20-week old males) were fasted overnight for ~12 h and then co-injected with glucose and CNO (glucose, 2 g/kg, i.p.; CNO, 3 mg/kg, i.p.), followed by the monitoring of plasma insulin levels.

Blood glucose levels were determined by using an automated blood glucose reader (Contour Next; Ascensia Diabetic Care). Plasma insulin levels were monitored by using an ELISA kit (Crystal Chem Inc.), following the manufacturer's instructions.

### Intravenous injections
In a subset of experiments, mice were subjected to isoflurane inhalation (Baxter Healthcare Corporation), followed by the opening of their abdominal cavity. Subsequently, mice were injected into the inferior vena cava with 100 µl of CNO solution in 0.9% saline (3 mg/kg). For control purposes, a subgroup of mice was injected via the same route with 100 µl of 0.9% saline. Five min after injections, livers were harvested and snap-frozen in liquid nitrogen.

### In vivo analysis of hepatic glucose fluxes
Male Hep-G12D mice and control mice (age: ~15 weeks) were obtained by i.v. AAV injections as described under "Generation of mutant mouse strains". Studies were initiated two weeks after virus treatment. Catheters were implanted under isoflurane anesthesia in the right jugular vein and left common carotid artery for infusion of tracers and sampling of blood, respectively[28,71]. Following the surgery, the animals were individually housed, and body weight was recorded. Five days after surgery, on the day of the study, mice were placed in bedded containers (at 7:00 am), and food was removed. Infusion lines were connected through a swivel system to the catheters of unrestrained conscious mice to allow freedom of movement. At 10:00 am (t = -150 min), a bolus of [6,6-D$_2$]-glucose (80 mg/kg) and D$_2$O (1.5 mg/kg) were given over a 40 min period. This was followed by a constant infusion of [6,6-D$_2$]-glucose (0.8 mg/kg/min) diluted in saline containing 4.5% D$_2$O that was maintained for the duration of the study. At t = −20 and 0 min, blood samples were taken to assess glucose concentrations and glucose enrichment. At t = 0 min, a bolus of CNO (3 mg/kg) was given via the jugular vein catheter. Blood samples were taken every 10 min for 50 min. Reconstituted red blood cells from a donor mouse were continuously infused (4 µl/min) for the duration of the study.

Glucose isotopomer distribution in arterial plasma was determined in the Vanderbilt Hormone Assay and Analytical Services Core using Agilent 5977 A MSD GC-MS according to the method of Antoniewicz et al.[72] and analyzed using isotopomer computational analysis software[73]. Glucose fluxes were assessed using non-steady-state equations (V$_d$ = 130 ml/kg)[74,75]. The contribution of gluconeogenesis was assessed as the enrichment ratio of C5/C2[72,76].

### Isolation and culture of primary mouse hepatocytes
Primary hepatocytes were isolated from livers of male mice (age: 12–15 weeks) by using a two-step collagenase perfusion protocol[77]. Hepatocytes (~0.7 × 10$^6$ cells per well) were cultured in collagen I-coated 6-well plates (Corning) in a 5% CO$_2$ incubator at 37 °C

(medium: DMEM containing 4.5 g glucose/l and 10% FBS). When cells were ~60–70% confluent, they were used for glucose production and other assays (see below).

### Measurement of glucose output from primary mouse hepatocytes
Primary mouse hepatocytes (0.7 × 10$^6$ cells per well) were cultured in 6-well plates (Corning) for 4–6 h at 37 °C. The culture medium consisted of phenol red-free DMEM containing 10% FBS and 4.5 g/l glucose. The medium was then replaced with fresh DMEM (phenol red-free) containing 1 g/l glucose. Following this step, hepatocytes were cultured overnight and then washed thoroughly with PBS. To stimulate glucose production, the medium was replaced with glucose- and phenol red-free DMEM supplemented with two gluconeogenic substrates, sodium lactate (20 mM) and sodium pyruvate (2 mM). Hepatocytes were then incubated at 37 °C for 5 hr in the presence of glucagon (100 nM), CNO (10 µM), and/or various pharmacological inhibitors. Subsequently, the culture medium was collected for the measurement of glucose concentrations using a glucose assay kit (Sigma). To measure total protein per well via the BCA method, hepatocytes were scraped off the wells with RIPA buffer containing proteinase inhibitor cocktail from Roche.

In a subset of experiments, primary hepatocytes prepared from Gna12$^{-/-}$ and WT control mice were initially transduced with an adenovirus (Ad-CMV-HA-G12D or Ad-CMV-eGFP; 1×10$^5$–5 × 10 infectious units/ml) for 4 h (medium: phenol red-free DMEM containing 10% FBS and 4.5 g/l glucose) in a 5% CO$_2$ incubator at 37 °C. After this step, the cells were washed thoroughly with PBS, and glucose output assays were performed as described in the previous paragraph.

### Measurement of glucose output from human HepG2 cells
HepG2 cells (ATCC, cat # HB-8065) were seeded into 12-well plates (Corning; cat. # #356500) at a density of 0.4 ×10$^6$ cells/well (medium: DMEM, low glucose [1 g/l]; Gibco). Six hr later, cells were transduced in the same medium with an adenovirus coding for G12D (Ad-CMV-HA-G12D) or the Ad-CMV-eGFP control virus (MOI: ~12 viral particles per cell). After a 2 hr incubation period, the medium was replaced with fresh low-glucose medium. On the following day, glucose output assays were carried out essentially as described under 'Hepatic glucose production studied with primary mouse hepatocytes'. HepG2 cells were incubated with gluconeogenic substrates (20 mM sodium lactate and 2 mM sodium pyruvate, respectively) for 3 h at 37 °C.

### Measurement of the activity of metabolically important liver enzymes
Hep-G12D and control mice (12-week-old males) were injected i.p. with either CNO (3 mg/kg) or saline. Thirty min later, the mice were euthanized, livers were harvested, and snap-frozen liver lysates were prepared for enzyme activity assays. Specifically, we monitored the enzymatic activities of glycogen phosphorylase, GSK3β, and ROCK. The formation of G6P was determined by using a colorimetric kit. The source of the assay kits used for these measurements are listed in Supplementary Table 1. Assays were carried out according to the guidelines provided by the manufacturers.

### Preparation of cDNA and RT-PCR analysis
Using standard molecular techniques, cDNAs were prepared from mouse primary hepatocytes or other mouse tissues (cell types) for qRT-PCR studies (see Supplementary Table 2 for primer sequences). Total RNA was extracted by using an RNA kit from Qiagen, following the manufacturer's protocol. qRT-PCR studies were carried using standard conditions[78]. Gene expression data were normalized relative to the expression of the β-actin gene.

## Knockdown of *GNA12* expression in HepG2 cells

HepG2 cells were cultured in 12-well plates (Corning Costar; cat. # 3513) for ~6 h at 37 °C in a 5% $CO_2$ incubator (medium: phenol-free DMEM containing 10% FBS and 4.5 g/l glucose). The medium was then replaced with fresh one, and HepG2 cells were cultured overnight. On the next day, cells were thoroughly washed and incubated with scrambled control siRNA or *GNA12* siRNA for 6 hr in Opti-MEM reduced serum medium (Thermo Fisher Scientific). Following the addition of fresh DMEM medium, cells were then cultured for an additional 24 h.

## Western blotting studies

Immunoblotting studies were carried out using mouse liver lysates or lysates from primary mouse hepatocytes. Immunoblotting studies were performed using standard procedures (~10 μg protein per lane). Immunoreactive proteins were visualized by using SuperSignal West Dura Chemiluminescent Substrate (Pierce). All antibody-related information is provided in Supplementary Table 1.

## A modified NanoBiT-G protein dissociation assay

CNO-induced G protein dissociation was measured by the NanoBiT-G protein dissociation assay[13] with modifications as described below. One day prior to transfection, HEK293A cells (Thermo Fisher Scientific, cat no. R70507) were seeded in 6-well culture plates at a density of 2 ×10⁵ cells/ml (medium: 2 ml DMEM (Nissui) supplemented with 5% fetal bovine serum (Gibco) and penicillin-streptomycin-glutamine; complete DMEM). To generate plasmids coding for G protein subunits useful for NanoBiT-G protein dissociation assays, the large fragment (LgBiT) of NanoBiT luciferase was inserted into the helical domain of $G\alpha_{12}$ or $G\alpha_{13}$ ($G\alpha_{12}$-LgBiT or $G\alpha_{13}$-LgBiT, respectively). The small fragment (SmBiT) of NanoBiT luciferase was fused to the N-terminus of $G\gamma_2$ containing the C68S point mutation[79] that prevents $G\gamma_2$ prenylation (SmBiT-$G\gamma_2$-CS). The modified $G\alpha_{12}$ (or $G\alpha_{13}$) and $G\gamma_2$ subunits were co-expressed with plasmids coding for untagged $G\beta_1$ and Ric8A[13], a G protein chaperone[80] (for detailed procedures regarding the generation of these various constructs, see ref. 13). Transfection solution was prepared by combining 6 μl (per well hereafter) of polyethyleneimine PEI MAX solution (1 mg/ml; Polysciences), 200 μl of Opti-MEM (Thermo Fisher Scientific), and a plasmid mixture consisting of 200 ng of the G12D construct (or empty plasmid for mock transfections), 100 ng Gα-LgBiT, 500 ng $G\beta_1$, 500 ng SmBiT-$G\gamma_2$-CS, and 100 ng of Ric8A plasmid DNA[13]. After a 1-day incubation period, the transfected cells were harvested with Dulbecco's PBS containing 0.5 mM EDTA, centrifuged, and suspended in 2 ml of HBSS containing 0.01 % bovine serum albumin (BSA; fatty acid-free grade; Serva) and 5 mM HEPES (pH 7.4) (assay buffer). The cell suspension was dispensed in a white 96-well plate at a volume of 80 μl per well and combined with 20 μl of 50 μM coelenterazine (Angene) diluted in assay buffer. After a 2 h incubation at room temperature, background luminescence was determined using a luminescence microplate reader (SpectraMax L, Molecular Devices), and defined amounts of CNO (20 μl; 6x of final concentrations) were added. Luminescence signals were measured 5–10 min after the addition of CNO and normalized to the initial counts. The resulting values (fold change) were further normalized to those of vehicle-treated samples and used to generate G protein dissociation-response curves. The G protein dissociation signals were fitted to a four-parameter sigmoidal concentration-response curve using Prism 9 software (GraphPad Prism),

## Flow cytometry

To determine which percentage of hepatocytes expressed the G12D receptor in Hep-G12D mice, we used flow cytometry to detect the presence of the HA epitope tag present at the extracellular N-terminus of G12D (Fig. 1a). We incubated hepatocytes isolated from control and Hep-G12D mice for 30 min on ice with an anti-HA antibody conjugated to Alexafluor-488. DNA content was quantified with DAPI (Thermo Scientific). All flow cytometry analyses were performed on a Beckman CytoFLEX S Flow Cytometer. The flowcytometry data were analyzed with FlowJo.

## cAMP assay

HEK293A cells were seeded into 6-well plates at a density of $2 \times 10^5$ cells/well in high-glucose DMEM (4.5 g glucose/l; Sigma) containing 10% FBS at 37 °C in a 5% $CO_2$ incubator. At 60-70% confluency, the cells were transfected with 1–2 μg/well of G12D-pcDNA3.1[22] using Lipofectamine (Thermo Fisher Scientific). The cells were then allowed to grow for an additional 24–48 h. Transfected cells were then washed with PBS, followed by the addition of high-glucose DMEM lacking FBS. Three hr later, cells were incubated with CNO (1 and 10 μM) or forskolin (8 and 40 μM) for 30 min at 37 °C. Subsequently, cells were lysed in 0.1 M HCl, and cAMP assays were performed using a cAMP ELISA kit according to the manufacturer's instructions (Cayman Chemical).

In an analogous fashion, primary mouse hepatocytes ($0.7 \times 10^6$ cells per well) prepared from control and Hep-G12D mice were cultured in 6-well plates (Corning) for 4-6 hr at 37 °C. The culture medium consisted of phenol red-free DMEM containing 10% FBS and 4.5 g glucose/l. The medium was then replaced with fresh DMEM (phenol red-free) containing 1 g/l glucose. Subsequently, hepatocytes were cultured overnight. Hepatocytes were then washed thoroughly with PBS and incubated with CNO or forskolin (10 μM each) for 30 min. After this incubation step, cells were processed for cAMP assays as described in the previous paragraph.

## IP₁ assay

Briefly, HEK293A cells ($0.5–1 \times 10^4$ cells/well) were transfected with 0.1–0.2 μg/well of plasmid DNA (G12D-pcDNA3.1 or GqD (hM3Dq)-pcDNA3.1) using Lipofectamine reagent (Thermo Fisher Scientific). Cells were cultured in 384-well plates in high-glucose DMEM (4.5 g glucose/l; Sigma) containing 10% FBS at 37 °C in a 5% $CO_2$ incubator. After transfection, the cells were allowed to grow for an additional 24 h. Cells were then washed with PBS, followed by the addition of high-glucose DMEM lacking FBS. Three hr later, cells were incubated with CNO (10 μM) for 30 min at 37 °C, followed by the measurement of intracellular IP₁ levels by using the HTRF IP-One Gq Detection Kit (Cisbio/Revvity).

## Preparation of human liver RNA and *GNA12* and *GNA13* expression analysis

Percutaneous liver biopsies of adult humans with MASLD ($n = 24$) were obtained in a clinical trial (clinicaltrials.gov NCT01792115) at baseline[81,82]. Participant characteristics are detailed in Supplementary Table 3. The study was approved by the NIDDK/NIAMS Institutional Review Board, and all subjects provided written informed consent.

A 2-5 mm sample from the liver biopsy specimen was flash-frozen at the bedside and archived at −80 °C. For analysis, samples were homogenized in TRIzol (Invitrogen), and total RNA was extracted using chloroform phase separation and a Qiagen kit. The quality of the extracted RNA was assessed using a NanoDrop spectrophotometer (Thermo Scientific) and Agilent 2100 Bioanalyzer. Samples were pooled (two pools) and run on an Illumina NovaSeq 6000 system with 150 bp paired-end sequencing. Reads were analyzed using Partek Flow. Reads underwent quality checking and were then trimmed (Pred score < 20). Reads were aligned to the human genome (GRCh38) using STAR version 2.7.3a with default settings. Genes were counted and annotated with Ensemble (release 100). Counts were normalized by size factors using the median-of-ratios method and corrected for batch effects using the general linear model. Gene expression data were obtained by using RStudio (version 2023.7.1 + 524). Possible correlations between *GNA12* or *GNA13* counts and parameters of insulin sensitivity and glucose homeostasis were assessed by using the Pearson correlation method.

## Determination of insulin sensitivity and glucose tolerance in humans

Plasma glucose and insulin measurements were obtained during the fasting state on the day on which liver biopsies were performed. The homeostatic model assessment for insulin resistance (HOMA-IR) was calculated as glucose [mg/dl] x insulin [mIU/l]/405[83]. An insulin-modified frequently-sampled intravenous glucose tolerance test (FS IVGTT) was performed 24-48 h prior to obtaining liver biopsy samples[84]. Minimal model analysis using the MINMOD software was used to calculate insulin sensitivity (Si)[85].

## Statistical analysis

Data are expressed as means ± s.e.m. for the indicated number of observations. Prior to performing specific statistical tests, we performed tests for normality and homogeneity of variance. Data were then tested for statistical significance by one- or two-way ANOVA, followed by the indicated posthoc tests, or by using a two-tailed unpaired Student's t-test, as appropriate. A $P$-value of less than 0.05 was considered statistically significant. The specific statistical tests that were used are indicated in the figure legends.

## Reporting summary

Further information on research design is available in the Nature Portfolio Reporting Summary linked to this article.

## Data availability

All raw data needed to reproduce the findings presented here can be found in the manuscript, figures or supplementary material. Any additional information is available from the corresponding author upon request. Source data are provided with this paper.

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

## Acknowledgements

This research was funded by the Intramural Research Program of the National Institute of Diabetes and Digestive and Kidney Diseases (NIDDK, NIH) and R01DK129946 (Y.B.K). We thank Kayo Sato, Shigeko Nakano, and Ayumi Inoue at Tohoku University for their assistance in preparing plasmids and carrying out the G protein signaling assays and Yuki Ono and Kaito Arai for helpful discussions. We are also grateful to Dr. Kyle D. Copps (Harvard Medical School, Dr. Morris M. White Lab) for his advice regarding some of the immunoblotting studies. A.I. was funded by KAKENHI JP21H04791, JP21H05113, JP21H0503, and JP24K21281 from the Japan Society for the Promotion of Science (JSPS), JP22ama121038 and JP22zf0127007 from the Japan Agency for Medical Research and Development (AMED), and JPMJFR215T, JPMJMS2023, and 22714181 from the Japan Science and Technology Agency (JST).

## Author contributions

S.P. and J.W. designed the study. S.P., D.H., O.M, Y.C., W-M.Y., A.M.W., Y.R., and A.I. carried out experiments and interpreted and analyzed experimental data. Y-B.K., R.D., and A.I. provided mutant mouse models and helpful advice. S.P. and J.W. wrote the manuscript.

## Funding

## Competing interests

The authors declare no competing interests.
