## [Transparent Peer Review file · Nature Communications]

G12/13-mediated signaling stimulates hepatic glucose production and has a major impact on whole body glucose homeostasis

Corresponding Author: Dr Jürgen Wess

Version 0:

Reviewer comments:

Reviewer #1

(Remarks to the Author)

The manuscript by Pittala et al. aims at understanding the role of hepatocyte G_{12}/G_{13} signaling in metabolic regulation. The authors use an elegant mouse model, in which G_{12}/G_{13} can be artificially activated through a recently developed new DREADD. The metabolic analysis of the mice is of very high quality, and the study is technically well done. The findings are interesting, but the analysis lacks depth, and the physiological and/or pathophysiological relevance of the findings remains unclear.

- Pittala et al. show that they are able to express the G12D DREADD in the liver. To be able to properly interpret the data, it is necessary to know whether expression in the liver is really restricted to hepatocytes and what percentage of hepatocytes expresses the designer receptor. Also, what is the level of expression compared to other GPCRs? This can at least be semi-quantitatively analyzed by comparing the levels of RNAs encoding the designer receptor to other endogenous G-protein-coupled receptors or by comparing receptor-mediated downstream signaling.

- G_{12} and G_{13} , the two G-proteins activated by G12D, have been shown to have in part independent functions. When looking at the manuscript data analyzing the contribution of G_{12} and G_{13} in the effects mediated by G12D activation, their individual roles remain unclear. While hyperglycemia induced by G12D activation *in vivo* appears to involve both G_{12} and G_{13} (Fig. 3), the *in vitro* experiments analyzing G12D-mediated glucose secretion suggest a primary involvement of G_{12} . This point required clarification. In particular, it would be interesting to test, if the two major effects mediated by G12D in the liver (glycogen breakdown and gluconeogenesis) involved either G_{12} or G_{13} , or both.

- How G_{12} and G_{13} mediate hepatocyte glycogenolysis and gluconeogenesis remains poorly understood. As this appears to be a potentially interesting new regulatory mechanism, more insight is required. How does ROCK activate JNK? In particular, it remains unclear how JNK regulates the activity of enzymes involved in glycogen breakdown and in gluconeogenesis. The authors show that activation of G12D leads to increased expression of genes encoding enzymes involved in gluconeogenesis (Fig. 5). However, strong effects on gluconeogenesis can already be seen 5 minutes after G12D activation (Fig. 2), which rather suggests a non-transcriptional mechanism. Here, certainly, a more in-depth analysis is required.

- While G12D expressed in the liver can, after activation by CNO, clearly induce a variety of interesting metabolic effects, it is important to test whether this can also be achieved by activation of an endogenous receptor under *in vivo* conditions.

- An important question unanswered is the physiological and/or pathophysiological significance of the described G_{12}/G_{13} -mediated regulation of hepatocyte function. Since the major *in vivo* findings are obtained with an artificial system, it would be very important to test the relevance of the regulation under more physiological conditions. In that respect, it is surprising that the authors could not find any phenotype in hepatocyte-specific $G\alpha_{12}/G\alpha_{13}$ -deficient mice.

Reviewer #2

(Remarks to the Author)

I would like to commend the authors for their great work in uncovering the role of G12/13 proteins in hepatic glucose production. Authors used a variety of up-to-date genetic models, and supported every result with multiple experiments that covers the spectrum from biochemical, to in-vitro and in-vivo, up to clinical.

However, I have some simple inquiries:

- 1- Authors used a significantly high concentration of CNO in the in-vitro experiment on hepatocytes. The half maximal effective concentration (EC50)= 8.1 nM for hM4Di (Jendryka et al., nature Scientific Reports 2019) or (EC50s = 16.7, 323, 17.4, 18.3, and 18.7 nM for PSMCs expressing hM1-5D receptors, respectively) (<https://www.caymanchem.com/product/16882/clozapine-n-oxide>), while authors used 10 micromolar concentration. I have concerns that CNO is no longer physiologically inert at such a high concentration. Do the authors have a justification for the chosen concentration? Do the authors have data that support the inertia of CNO at 10 micromolar?
- 2- The most prominent and significant glucose changes in in-vivo experiments were at the 5 minutes time point, while (Jendryka et al., nature Scientific Reports 2019) showed that CNO needs at least 15 minutes after intraperitoneal injection in mice to start action.
- 3- Authors showed that glucose changes in vivo were fast (within minutes) after CNO injection, and they think that these effects come from G12/13-ROCK-JNK-Gluconeogenesis/Glycogenolysis proteins expression pathway. I think it will not cause abrupt elevations in glucose (as shown in this article). Glucagon, for example, whose receptor is Gs coupled, activates cAMP and then induces phosphorylation changes in glycogenolytic/gluconeogenetic enzymes, and even this fast mechanism needs minutes to take action. Maybe the authors' suggested pathway play a role in long-term homeostasis.
- 4- Authors said: 'HepG12/G13 KO mice maintained on a HFD for 8 weeks did not differ from the corresponding control WT mice in fed and fasting blood glucose and plasma insulin levels, glucose tolerance, and insulin sensitivity (Extended Data Fig. 7c-e). These somewhat surprising observations suggest that compensatory hepatic signaling pathways (e.g., signaling via other classes of 18 heterotrimeric G proteins such as Gs or Gq/11)'
Did the authors investigate this possible compensatory mechanism?
- 5- Why did the authors use hepatocyte specific TBG as marker for hepatocyte to generate the genetic model? Why did not they use albumin like commonly done?
- 6- It would be better to provide supportive immunohistochemistry images for the tissues to prove that the G12D is only expressed in liver and specifically in hepatocytes.
- 7- Authors showed in fig.1b western blot data about the specific expression of G12 in liver, do the authors have western blot or histology data for the Ga13 in the established model in the different tissues? Because they also assume that Ga13 is knocked down in their model.
- 8- In fig.1F, why is the baseline blood glucose high even though the insulin tolerance test is performed after fasting.
- 9- Fig.3b, why is not there a Gna12^{-/-} Gna13^{fl/fl} with TBG-eGFP and TBG-Cre to know the role of Ga13 alone.
- 10- Fig.c7, the authors suppose in the graphical abstract that JNK is a downstream of ROCK, but they did not provide experimental evidence for that.

Reviewer #3

(Remarks to the Author)

This is an interesting and solid paper providing evidence that G12/13 signaling regulates hepatic glucose homeostasis.

SPECIFIC CONCERNS:

1. Given that the chemogenetic construct is modified it will be important to provide negative data for coupling to Gq/i/s-family G proteins (e.g. via BRET)

Version 1:

Reviewer comments:

Reviewer #1

(Remarks to the Author)

The authors did an excellent job of addressing the points I had raised in my review.

Reviewer #2

(Remarks to the Author)

Reviewer #3

(Remarks to the Author)

My concerns have been fully addressed.

General response by the authors

We would like to thank the three reviewers for their constructive comments and helpful suggestions which have further improved the impact and quality of our manuscript.

Please note that we carried out many additional experiments to address the various comments raised by the reviewers. The additional work that we performed has led to the incorporation of several new figure panels and 10 (!) additional Supplementary Figures into the revised version of the manuscript.

Reviewer #1

COMMENT

The manuscript by Pittala et al. aims at understanding the role of hepatocyte G₁₂/G₁₃ signaling in metabolic regulation. The authors use an elegant mouse model, in which G₁₂/G₁₃ can be artificially activated through a recently developed new DREADD. The metabolic analysis of the mice is of very high quality, and the study is technically well done. The findings are interesting, but the analysis lacks depth, and the physiological and/or pathophysiological relevance of the findings remains unclear.

OUR RESPONSE

Thank you for your very positive comments. As requested, we carried out additional experiments to add depth to the manuscript and to address the physiological and/or pathophysiological relevance of our findings (see below for details).

COMMENT

Pittala et al. show that they are able to express the G12D DREADD in the liver. To be able to properly interpret the data, it is necessary to know whether expression in the liver is really restricted to hepatocytes and what percentage of hepatocytes expresses the designer receptor.

OUR RESPONSE

To address this issue, we separated mouse hepatocytes from other liver cells (Kupffer cells, stellate cells, etc.). Western blotting studies confirmed that the G12D designer receptor was expressed only in hepatocytes and not in other cell types of the liver prepared from Hep-G12D mice (Supplementary Fig. 3) (new).

To determine which percentage of hepatocytes expressed the G12D receptor, we used flow cytometry analysis to detect the presence of the HA epitope tag present at the extracellular N-terminus of G12D (Fig. 1a) (see Methods for details). This analysis showed that 55 ±8% of purified hepatocytes prepared from Hep-G12D mice expressed the G12D construct (mean ± s.e.m.; n=3). In contrast, no significant HA signal was detected with hepatocytes derived from control littermates (n=3).

Appropriate changes have been made at the beginning of the Results section of the revised manuscript (page 8 bottom, and page 9, top).

COMMENT

Also, what is the level of expression compared to other GPCRs? This can at least be semi-quantitatively analyzed by comparing the levels of RNAs encoding the designer receptor to other

endogenous G-protein-coupled receptors or by comparing receptor-mediated downstream signaling.

OUR RESPONSE

To address this issue, we used qRT-PCR technology to compare the hepatic expression levels of the G12D designer receptor with those of several GPCRs known to be endogenously expressed in mouse hepatocytes. Specifically, we determined transcript levels of the following GPCRs (gene names in parentheses): glucagon receptor (*Gcgr*), V_{1A} vasopressin receptor (*Avpr1a*), sphingosine 1-phosphate receptor subtype 1 (*Slpr1*), and Gpr91 (*Sucnr1*). Using RNA prepared from Hep-G12D mice, we found that the G12D designer receptor was expressed at similar levels as the endogenously expressed glucagon receptor (Supplementary Fig. 4) (new) (note that lower ΔC_t values correspond to higher transcript levels). Moreover, hepatic *G12D* mRNA levels were only 2-4-fold higher than the corresponding *Slpr1* and *Sucnr1* transcript levels. These observations indicated that Hep-G12D mice express G12D at levels similar or close to those of GPCRs that are endogenously expressed by the liver. Absolute C_t values were (means \pm s.e.m.; n=3): *G12D*, 18.7 \pm 0.1; *Gcgr*: 18.5 \pm 0.1; *Avpr1a*, 24.5 \pm 0.1; *Slpr1*, 20.9 \pm 0.2; and *Gpr91*, 19.8 \pm 0.1.

Appropriate changes have been made in the revised manuscript (page 9, top).

COMMENT

G₁₂ and G₁₃, the two G-proteins activated by G12D, have been shown to have in part independent functions. When looking at the manuscript data analyzing the contribution of G₁₂ and G₁₃ in the effects mediated by G12D activation, their individual roles remain unclear. While hyperglycemia induced by G12D activation *in vivo* appears to involve both G₁₂ and G₁₃ (Fig. 3), the *in vitro* experiments analyzing G12D-mediated glucose secretion suggest a primary involvement of G₁₂. This point required clarification. In particular, it would be interesting to test, if the two major effects mediated by G12D in the liver (glycogen breakdown and gluconeogenesis) involved either G₁₂ or G₁₃, or both.

OUR RESPONSE

To address this issue, we studied hepatic glycogenolysis and gluconeogenesis in the following 5 groups of mice:

1. WT mice
2. Hep-G12D mice
3. Hep-G12D mice lacking G α_{12} (whole body KO)
4. Hep-G12D mice lacking G α_{13} in hepatocytes
5. Hep-G12D mice deficient in G α_{12} (whole body KO) and lacking G α_{13} in hepatocytes

We then carried out the experiments described in the following paragraph (page 13 of the revised manuscript): "Both G₁₂ and G₁₃ are required for G12D-mediated hepatic glycogenolysis and gluconeogenesis"

We explored whether G12D-mediated increases in the rates of glycogen breakdown and gluconeogenesis involved either G₁₂ or G₁₃ signaling, or both G protein subtypes. To address this issue, we first studied hepatic glycogenolysis using five different mouse strains (WT mice, Hep-G12D mice, Hep-G12D mice deficient in G α_{12} (whole body), Hep-G12D mice lacking G α_{13} in hepatocytes only, and Hep-G12D mice deficient in G α_{12} (whole body) and lacking G α_{13} in

hepatocytes only) (Supplementary Fig. 9) (new). Livers were harvested 30 min after CNO treatment (3 mg/kg, i.p.). We then used liver lysates to measure the activities of hepatic glycogen phosphorylase and Gsk3 β , two enzymes that play central roles in promoting glycogenolysis. As expected, CNO treatment of Hep-G12D mice caused a pronounced increase in the activities of both enzymes (Supplementary Fig. 9) (new). These effects were strongly attenuated or abolished in CNO-treated Hep-G12D mice deficient in either G α_{12} or lacking G α_{13} in hepatocytes, indicating that both G $_{12}$ and G $_{13}$ signaling contribute to the G12D-mediated increase in hepatic glycogenolysis (Supplementary Fig. 9) (new).

To explore the role of G $_{12}$ and G $_{13}$ signaling in stimulating hepatic gluconeogenesis, we prepared hepatocytes from the same five groups of mice described in the previous paragraph. We then incubated hepatocytes with CNO (10 μ M) and high concentrations of two major gluconeogenic substrates (sodium lactate and sodium pyruvate) for 4 hr, followed by the measurement of glucose release. Expectedly, CNO treatment of hepatocytes prepared from Hep-G12D mice strongly stimulated glucose secretion (Supplementary Fig. 10) (new). This response was greatly reduced in hepatocytes obtained from Hep-G12D mice deficient in either G α_{12} or lacking G α_{13} in hepatocytes (Supplementary Fig. 10) (new). This observation indicates that both G $_{12}$ and G $_{13}$ signaling contribute to the G12D-mediated stimulation of hepatic gluconeogenesis.

These findings are in good agreement with the *in vivo* data shown in Fig. 3. These new data are described in the revised manuscript (pages 13, 14 (top), and 26 (last sentence of the penultimate paragraph)).

In Fig 6d, we stimulated sphingosine 1-phosphate (S1P) subtype 1 receptors (S1PR1) that are endogenously expressed by hepatocytes with ponesimod, an S1PR1-selective agonist. While ponesimod led to a robust secretion of glucose from primary hepatocytes derived from WT mice, this response was virtually abolished in primary hepatocytes obtained from mice lacking G α_{12} . This observation is consistent with the recent finding that activation of S1PR1 can recruit G $_{12}$, but not G $_{13}$, with high efficacy (Avet *et al.*, 2022). We added this later statement to the main text of the revised manuscript (page 21, last sentence of the penultimate paragraph).

COMMENT

How G $_{12}$ and G $_{13}$ mediate hepatocyte glycogenolysis and gluconeogenesis remains poorly understood. As this appears to be a potentially interesting new regulatory mechanism, more insight is required.

How does ROCK activate JNK?

OUR RESPONSE

It has been reported that stimulation of ROCK1 can promote the phosphorylation and activation of JNK in different cell types (Marinissen *et al.*, 2004; Ongusaha *et al.*, 2008; Schofield and Bernard, 2013). Previous work has shown that ROCK1 does not phosphorylate JNK directly but that this phosphorylation event is mediated indirectly by other kinases downstream of ROCK1, whose molecular identity remains to be identified (Marinissen *et al.*, 2004). Thus, it is likely that a similar mechanism is operative in mouse hepatocytes.

We discussed this issue with some of the top experts in the ROCK/JNK field.

This information has been included in the revised manuscript (page 20, bottom paragraph).

COMMENT

In particular, it remains unclear how JNK regulates the activity of enzymes involved in glycogen breakdown and in gluconeogenesis.

OUR RESPONSE

To address this issue, we carried out a series of Western blotting experiments exploring the phosphorylation status of key signaling proteins, enzymes, and transcription factors regulating hepatic glucose fluxes.

JNK is known to promote the dephosphorylation of pFoxO1, resulting in the accumulation of the non-phosphorylated form of FoxO1 in the nucleus (Deng et al., 2011; Wang et al., 2005; Weng et al., 2016). Nuclear FoxO1 acts as a transcription factor that strongly promotes the expression of gluconeogenic genes including *G6pc* and *Pepck* (Lin and Accili, 2011). To investigate whether activation of G12D promotes the dephosphorylation of pFoxO1 in hepatocytes, we injected Hep-G12D mice and control littermates with CNO (3 mg/kg, i.p.). Thirty min later, mice were euthanized, and liver lysates were prepared and subjected to Western blotting studies. We found that CNO treatment of Hep-G12D mice caused a marked reduction in hepatic pFoxO1 levels, as compared to CNO-treated control littermates (Supplementary Fig. 11) (new). This observation is consistent with a model in which G12D-mediated signaling promotes the dephosphorylation of hepatic pFoxO1, resulting in a prolonged increase in hepatic gluconeogenesis. In agreement with this concept, the expression levels of the rate-limiting enzymes involved in the process of gluconeogenesis (glucose 6-phosphatase [*G6p*], fructose bisphosphatase [*Fbp*], and phosphoenolpyruvate carboxykinase 1 [*Pck1*]) were markedly increased in liver tissue of Hep-G12D mice prepared 30 min after CNO injection (3 mg/kg, i.p.) (Fig. 5e).

Under the same experimental conditions, the expression of pAkt, a key signaling hub in the insulin signaling pathway, was only slightly elevated in Hep-G12D mice (Supplementary Fig. 11) (new), most likely due to enhanced insulin secretion following CNO-induced hyperglycemia (see Fig. 1h).

As shown in Fig. 2f, CNO treatment of G12D mice stimulated gluconeogenesis and glycogen breakdown already after 5 min, indicative of the involvement of non-transcriptional processes. To explore the nature of such potential non-transcriptional mechanisms, we studied the phosphorylation status of several signaling proteins known to regulate hepatic glucose fluxes. Specifically, we injected Hep-G12D mice i.v. with either saline or CNO (3 mg/kg). Five min later, mice were euthanized, and liver lysates were prepared and subjected to Western blotting studies. We found that CNO-mediated activation of G12D led to the rapid inhibitory phosphorylation of liver glycogen synthase at position S641 and the activating phosphorylation of liver glycogen phosphorylase (PYGL) at position S15, respectively (Supplementary Fig. 12a, b) (new). These rapid phosphorylation events provide a molecular basis for the quick onset of G12D-mediated glycogen breakdown.

Moreover, while pJNK was barely detectable in livers from saline-treated Hep-G12D mice, CNO treatment of Hep-G12D mice led to a very robust increase in hepatic pJNK formation (Supplementary Fig. 12a, b) (new). CNO-mediated rapid stimulation of G12D had only a small or no significant effect on the phosphorylation status of Akt and FoxO1, respectively, two key components of the insulin receptor signaling cascade (Supplementary Fig. 12a, c) (new). We also found that CNO treatment of Hep-G12D mice led to a significant increase in the phosphorylation of hepatic Irs1 at S307 (Supplementary Fig. 12d) (new). The formation of pIrs1 (S307) has been shown to be mediated by activated JNK (pJNK) and interferes with insulin receptor signaling (Bogoyevitch and Kobe, 2006; Hirosumi et al., 2002). Taken together, our data support the concept that G12D-mediated activation of JNK plays a central role in the rapid changes in hepatic glucose fluxes observed with CNO-treated Hep-G12D mice.

These new data have been incorporated into the text of the revised manuscript (Results, pages 16 and 17; Discussion, pages 27 (bottom half) and 28).

We are aware of the fact that these new observations do not provide a compressive explanation of how JNK regulates the activity of enzymes involved in glycogen breakdown and gluconeogenesis. As reviewed recently (Zeke *et al.*, 2016), pJNK can act on more than 100 different cellular substrates including various protein phosphatases and kinases. Clearly, a comprehensive analysis of the dozens of cellular targets of pJNK is beyond the scope of the present study. We are planning to systematically explore the cellular substrates and signaling mechanisms through which G_{12/13}-dependent hepatic JNK activation affects hepatic glucose metabolism in a detailed follow-up study (see Discussion, page 28, center).

COMMENT

The authors show that activation of G12D leads to increased expression of genes encoding enzymes involved in gluconeogenesis (Fig. 5). However, strong effects on gluconeogenesis can already be seen 5 minutes after G12D activation (Fig. 2), which rather suggests a non-transcriptional mechanism. Here, certainly, a more in-depth analysis is required.

OUR RESPONSE

As discussed in the past, rapid hepatic gluconeogenesis (within minutes) can be caused by many transcription-independent mechanisms (Barroso *et al.*, 2024; Burgess, 2015; Exton *et al.*, 1970; Lin and Accili, 2011), including the allosteric inhibition of key regulatory glycolytic enzymes, the activation of pathways that alter the availability of gluconeogenetic substrates (lactate, amino acids, glycerol, etc.), the cytoplasmic redox state, or the activity of various phosphodiesterases and other enzymes or signaling pathways that regulate signaling via hepatic glucagon and insulin receptors (Barroso *et al.*, 2024; Exton *et al.*, 1970; Lin and Accili, 2011). Additional factors that could contribute to transcription-independent increases in gluconeogenesis include changes in posttranslational modifications of existing signaling complexes or altered function of different components of the mitochondrial electron transport chain (Barroso *et al.*, 2024; Exton *et al.*, 1970; Lin and Accili, 2011).

We observed that CNO treatment of Hep-G12D mice led to a very rapid (with 5 min) and pronounced increase in hepatic pJNK formation (Supplementary Fig. 12a, b) (new). Since pJNK can act on more than 100 different cellular substrates including various protein phosphatases and kinases (Zeke *et al.*, 2016), it is likely that the rapid G12D-mediated formation of pJNK makes a key contribution to the quick increase in hepatic glucose output observed with CNO-treated Hep-G12D mice (Fig. 2f). CNO treatment of Hep-G12D mice also led to a rapid increase in the phosphorylation of Irs1 at Ser307 (Supplementary Fig. 12d) (new). Previous studies have shown that this phosphorylation step is mediated by activated JNK (pJNK) (Bogoyevitch and Kobe, 2006; Hirosumi *et al.*, 2002), resulting in impaired insulin receptor signaling. Taken together, these data support the concept that G12D-mediated activation of JNK plays a central role in the rapid stimulation of gluconeogenesis observed with CNO-treated Hep-G12D mice. These new data have been incorporated into the text of the revised manuscript (Results, page 16, first two paragraphs, and page 17, lines 1-13; Discussion, page 28, center).

In addition, we wanted to exclude the possibility that G12D-mediated increases in hepatic cAMP levels underly the observed rapid increase in the rate of hepatic gluconeogenesis. Glucagon and other agents that stimulate the production of cAMP in hepatocytes lead to rapid increases in HGP (Exton, 1987; Lin and Accili, 2011). Previous studies have shown that receptor-induced G_{12/13} signaling can increase cytoplasmic cAMP levels via activation of

adenylyl cyclase isoform 7 (AC7) in certain cell types (Jiang et al., 2008; Jiang et al., 2013). To explore whether this pathway is operative in mouse hepatocytes, we treated primary hepatocytes prepared from Hep-G12D mice with CNO (10 μ M), followed by the monitoring of intracellular cAMP levels. We found that CNO treatment of G12D-expressing hepatocytes had no significant effect on cytoplasmic cAMP levels (Supplementary Fig. 13) (new). In contrast, forskolin (10 μ M; positive control) induced a very robust cAMP response in these cells (Supplementary Fig. 13) (new). These data indicate that changes in intracellular cAMP levels do not play a role in the rapid changes in hepatic glucose production observed after simulation of hepatic G_{12/13} signaling.

These new data have been incorporated into the text of the revised manuscript (page 19, bottom paragraph).

As already discussed above, these new data do not provide a compressive explanation of how G12D-mediated JNK activation results in rapid hepatic gluconeogenesis. Since pJNK can act on dozens of cellular substrates including various protein phosphatases and kinases (Zeke *et al.*, 2016), we are planning to systematically explore the cellular targets and signaling mechanisms through which G_{12/13}-dependent hepatic JNK activation affects rapid hepatic gluconeogenesis in a detailed follow-up study (see Discussion, page 28, center).

COMMENT

While G12D expressed in the liver can, after activation by CNO, clearly induce a variety of interesting metabolic effects, it is important to test whether this can also be achieved by activation of an endogenous receptor under *in vivo* conditions.

OUR RESPONSE

Previous studies have shown that the S1P₁ receptor subtype (S1PR1), which is highly expressed in the liver (Regard *et al.*, 2008), can couple to G proteins of the G_{12/13} family with high efficacy (Avet *et al.*, 2022). To explore the effect of activating S1PR1s endogenously expressed by hepatocytes on glucose secretion, we acutely treated hepatocytes isolated from WT and *Gna12*^{-/-} mice with ponesimod, a selective S1PR1 agonist (Piali *et al.*, 2011). We found that ponesimod enhanced glucose release from mouse hepatocytes in a G₁₂-dependent fashion (Fig. 6d). This observation is consistent with the recent finding that activation of S1PR1s can recruit G₁₂, but not G₁₃, with high efficacy (Avet *et al.*, 2022). These *in vitro* data were presented in the original version of the manuscript.

To address the issue raised by the reviewer ("activation of an endogenous receptor under *in vivo* conditions"), we injected ponesimod (20 mg/kg, i.p.) into WT mice, whole body G α ₁₂ KO mice (G12 KO mice), and G12 KO mice lacking G α ₁₃ selectively in hepatocytes (Hep-G12/13 KO mice). We then monitored changes in blood glucose levels over a 90 min time period. Interestingly, ponesimod treatment of WT mice resulted in a mild but significant increase in blood glucose levels, as compared to saline-treated WT mice (Supplementary Fig. 15a) (new). This effect was absent in both G12 KO and Hep-G12/13 KO mice (Supplementary Fig. 15b, c) (new). Thus, the outcome of these *in vivo* studies is consistent with the *in vitro* data obtained with isolated hepatocytes (Fig. 6d).

We modified the text of the revised manuscript accordingly (page 21, bottom paragraph).

COMMENT

An important question unanswered is the physiological and/or pathophysiological significance of the described G_{12/13}-mediated regulation of hepatocyte function. Since the major *in vivo* findings

are obtained with an artificial system, it would be very important to test the relevance of the regulation under more physiological conditions. In that respect, it is surprising that the authors could not find any phenotype in hepatocyte-specific $G\alpha_{12}/G\alpha_{13}$ -deficient mice.

OUR RESPONSE

We now provide clear evidence indicating that hepatic $G_{12/13}$ signaling is of physiological relevance. In Fig. 7 of the original manuscript, we showed that fasted WT mice displayed a significant upregulation of *Gna12* and *Rock1* expression in the liver. At the same time, hepatic ROCK activity was also significantly enhanced in fasted WT mice (Fig. 7, original manuscript). These data strongly suggest that fasting promotes G_{12} -ROCK signaling, which in turn is predicted to stimulate HGP.

To explore whether the fasting-induced increases in *Rock1* expression and ROCK activity were mediated by enhanced hepatic $G_{12/13}$ signaling, we carried out analogous studies with fasted Hep-G12/13 KO mice (*Gna12*^{-/-} *Gna13* *fl/fl* mice treated with the AAV-TGB-Cre virus). Strikingly, the fasting-induced increases in hepatic *Rock1* expression and ROCK activity were abolished in Hep-G12/13 KO mice (Fig. 7b-d) (new). This finding strongly supports the concept that enhanced activity of the $G_{12/13}$ -ROCK1 signaling cascade contributes to the maintenance of euglycemia under fasting conditions.

These new data have been incorporated into the revised version of the manuscript (Results, page 23, center; Discussion, page 29, bottom paragraph).

We also measured fasting blood glucose levels of WT and Hep-G12/13 KO mice (males) that had been maintained on a high-fat diet (HFD) for an extended period of time (16 weeks). Under these experimental conditions, Hep-G12/13 KO mice showed significantly lower blood glucose levels than HFD control mice (Supplementary Fig. 16f) (new), indicating that hepatic $G_{12/13}$ signaling affects fasting blood glucose levels after prolonged consumption of a high-calorie diet. These new data have been incorporated into the revised version of the manuscript (Results, page 24, top paragraph; Discussion, page 29, bottom paragraph).

In addition, as discussed above, we found that activation of the S1PR1 receptor subtype, which is endogenously expressed by hepatocytes, stimulated glucose release from mouse hepatocytes in a G_{12} -dependent fashion (Fig. 6d). We obtained similar results when we injected ponesimod, an S1PR1 agonist, into WT mice, whole body $G\alpha_{12}$ KO mice (G12 KO mice), and G12 KO mice lacking $G\alpha_{13}$ selectively in hepatocytes (Hep-G12/13 KO mice) (Supplementary Fig. 15) (new).

We modified the text of the revised manuscript accordingly (page 21, bottom paragraph).

In this context, it is also important to note that studies with human liver samples indicated that hepatic *GNAI2* (encoding $G\alpha_{12}$) expression levels positively correlated with indices of insulin resistance and impaired glucose homeostasis, consistent with a potential pathophysiological role of enhanced hepatic $G_{12/13}$ signaling (Supplementary Fig. 17 in the revised manuscript).

Taken together, these studies strongly suggest that hepatic $G_{12/13}$ signaling plays a physiological and potential pathophysiological role in maintaining euglycemia.

NOTE: To highlight the physiological relevance of hepatic $G_{12/13}$ signaling, we added an additional paragraph to the Discussion section (page 29, bottom, and page 30, top).

Reviewer #2 (Remarks to the Author):

I would like to commend the authors for their great work in uncovering the role of G12/13 proteins in hepatic glucose production. Authors used a variety of up-to-date genetic models, and supported every result with multiple experiments that covers the spectrum from biochemical, to in-vitro and in-vivo, up to clinical. However, I have some simple inquiries:

COMMENT

Thank you for your very positive comments.

COMMENT

1- Authors used a significantly high concentration of CNO in the in-vitro experiment on hepatocytes. The half maximal effective concentration (EC50)= 8.1 nM for hM4Di(Jendryka et al., nature Scientific Reports 2019) or (EC50s = 16.7, 323, 17.4, 18.3, and 18.7 nM for PSMCs expressing hM1-5D receptors, respectively) (<https://www.caymanchem.com/product/16882/clozapine-n-oxide>), while authors used 10 micromolars concentration. I have concerns that CNO is no longer physiologically inert at such a high concentration. Do the authors have a justification for the chosen concentration? Do the authors have data that support the inertia of CNO at 10 micromolars?

OUR RESPONSE

Most studies employing CNO for in vitro work have used CNO at a concentration of 10 μ M to ensure maximum DREADD occupancy. To the best of our knowledge, we are not aware of any reports that this concentration of CNO had significant non-specific effects on the function of non-DREADD expressing control cells (Google scholar lists nearly 300 papers that used CNO at 10 μ M in vitro). In the past, we used 10 μ M of CNO to explore the outcome of activating a G_q DREADD (Li et al., 2013), a G_s DREADD (Akhmedov et al., 2017), or a G_i DREADD (Rossi et al., 2018) expressed in mouse hepatocytes. In these studies, treatment of control cells (non-DREADD-expressing hepatocytes) with 10 μ M of CNO had no detectable effect on hepatocyte function including glucose output.

COMMENT

2- The most prominent and significant glucose changes in in-vivo experiments were at the 5 minutes time point, while (Jendryka et al., nature Scientific Reports 2019) showed that CNO needs at least 15 minutes after intraperitoneal injection in mice to start action.

OUR RESPONSE

As stated by the reviewer, a pronounced increase in blood glucose levels was already observed 5 min after CNO treatment (Fig. 2). In this experiment, CNO was administered intravenously (i.v.) (see legend to Fig. 2), resulting in the rapid delivery of high concentrations of CNO to the liver. Thus, it is likely that the different routes of CNO application (i.v. (current study) versus i.p. (Jendryka et al.)) are responsible for the different time courses of CNO actions.

COMMENT

3- Authors showed that glucose changes in vivo were fast (within minutes) after CNO injection, and they think that these effects come from G12/13-ROCK-JNK-Gluconeogenesis/Glycogenolysis proteins expression pathway. I think it will not cause abrupt elevations in glucose (as shown in this article). Glucagon, for example, whose receptor is G_s

coupled, activates cAMP and then induces phosphorylation changes in glycogenolytic/gluconeogenic enzymes, and even this fast mechanism needs minutes to take action. Maybe the authors' suggested pathway play a role in long-term homeostasis.

OUR RESPONSE

We agree with the reviewer that the hepatic signaling cascade we characterized in this study is likely to play a role in the prolonged stimulation of hepatic glucose release.

However, it has been reported that rapid hepatic glycogenolysis and gluconeogenesis can occur by many transcription-independent mechanisms (Barroso *et al.*, 2024; Burgess, 2015; Exton *et al.*, 1970; Lin and Accili, 2011). Importantly, processes that lead to the allosteric inhibition or activation of key enzymes regulating glycogenolysis and gluconeogenesis can increase the rate of HGP rapidly (Barroso *et al.*, 2024; Burgess, 2015; Exton *et al.*, 1970; Lin and Accili, 2011).

As shown in Fig. 2f, CNO treatment of G12D mice stimulated gluconeogenesis and glycogen breakdown already after 5 min, indicative of the involvement of non-transcriptional processes. To explore the nature of such potential non-transcriptional mechanisms, we studied the phosphorylation status of several signaling proteins known to regulate hepatic glucose fluxes. Specifically, we injected Hep-G12D mice i.v. with either saline or CNO (3 mg/kg). Five min later, mice were euthanized, and liver lysates were prepared and subjected to Western blotting studies. We found that CNO-mediated activation of G12D led to the rapid inhibitory phosphorylation of liver glycogen synthase at position S641 and the activating phosphorylation of liver glycogen phosphorylase (PYGL) at position S15, respectively (Supplementary Fig. 12a, b) (new). These rapid phosphorylation events provide a molecular basis for the quick onset of G12D-mediated glycogen breakdown.

Moreover, we observed that CNO treatment of Hep-G12D mice led to a very rapid (with 5 min) and pronounced increase in hepatic pJNK formation (Supplementary Fig. 12a, b) (new). Since pJNK can act on more than 100 different cellular substrates including various protein phosphatases and kinases (Zeke *et al.*, 2016), it is likely that the rapid G12D-mediated formation of pJNK makes a key contribution to the quick increase in hepatic glucose output observed with CNO-treated Hep-G12D mice (Fig. 2f). CNO treatment of Hep-G12D mice also led to a rapid increase in the phosphorylation of Irs1 at Ser307 (Supplementary Fig. 12d) (new). Previous studies have shown that this phosphorylation step is mediated by activated JNK (pJNK) (Bogoyevitch and Kobe, 2006; Hirosumi *et al.*, 2002), resulting in impaired insulin receptor signaling. Taken together, these data support the concept that G12D-mediated activation of JNK plays a central role in the rapid stimulation of gluconeogenesis observed with CNO-treated Hep-G12D mice.

These new data have been incorporated into the text of the revised manuscript (Results, pages 16 and 17 (top half); Discussion, pages 27 and 28).

In addition, we wanted to exclude the possibility that G12D-mediated increases in hepatic cAMP levels underly the observed rapid increase in the rate of hepatic gluconeogenesis. Glucagon and other agents that stimulate the production of cAMP in hepatocytes lead to rapid increases in HGP (Exton, 1987; Lin and Accili, 2011). Previous studies have shown that receptor-induced G_{12/13} signaling can increase cytoplasmic cAMP levels via activation of adenylyl cyclase isoform 7 (AC7) in certain cell types (Jiang *et al.*, 2008; Jiang *et al.*, 2013). To explore whether this pathway is operative in mouse hepatocytes, we treated primary hepatocytes prepared from Hep-G12D mice with CNO (10 μ M), followed by the monitoring of intracellular cAMP levels. We found that CNO treatment (of G12D-expressing hepatocytes had no significant effect on cytoplasmic cAMP levels (Supplementary Fig. 13) (new). In contrast, forskolin (10

μM ; positive control) induced a very robust cAMP response in these cells (Supplementary Fig. 13) (new). These data indicate that changes in intracellular cAMP levels do not play a role in the rapid changes in hepatic glucose production observed after simulation of hepatic $G_{12/13}$ signaling.

These new data have been incorporated into the text of the revised manuscript (page 19, bottom).

We are aware of the fact that these new observations do not provide a compressive explanation of how JNK regulates the activity of enzymes involved in stimulating glycogen breakdown and in gluconeogenesis. As reviewed recently (Zeke *et al.*, 2016), pJNK can act on more than 100 different cellular substrates including various protein phosphatases and kinases. Clearly, a comprehensive analysis of the dozens of cellular targets of pJNK is beyond the scope of the present study. We are planning to systematically explore the cellular substrates and signaling mechanisms through which $G_{12/13}$ -dependent hepatic JNK activation affects hepatic glucose metabolism in a detailed follow-up study (see Discussion, page 28, center).

COMMENT

4- Authors said: `HepG12/G13 KO mice maintained on a HFD for 8 weeks did not differ from the corresponding control WT mice in fed and fasting blood glucose and plasma insulin levels, glucose tolerance, and insulin sensitivity (Supplementary Fig. 7c-e). These somewhat surprising observations suggest that compensatory hepatic signaling pathways (e.g., signaling via other classes of 18 heterotrimeric G proteins such as G_s or $G_{q/11}$)`

Did the authors investigate this possible compensatory mechanism?

OUR RESPONSE

We previously demonstrated that activation of hepatocyte G_s or $G_{q/11}$ signaling results in robust increases in blood glucose levels (Akhmedov *et al.*, 2017; Li *et al.*, 2013). To explore whether the hepatic expression levels of $G\alpha_s$ and $G\alpha_{q/11}$ were altered in HFD Hep-G12/13 KO mice, we carried out Western blotting studies using $G\alpha_s$ - and $G\alpha_{q/11}$ -specific antibodies. We found that hepatic $G\alpha_s$ and $G\alpha_{q/11}$ levels were similar in Hep-G12/13 KO and control mice consuming a HFD (Supplementary Fig. 16g) (new), indicating that the lack of hepatic $G_{12/13}$ signaling has no effect on the hepatic expression levels of $G\alpha_s$ and $G\alpha_{q/11}$ (see Results, page 24 (center paragraph) in the revised version).

Please note: We carried out additional metabolic studies with mice (males) that had been maintained on a HFD for a prolonged period of time (16 weeks). Under these experimental conditions, Hep-G12/13 KO mice showed significantly lower fasting blood glucose levels than their corresponding HFD control mice (Supplementary Fig. 16f) (new), indicating that hepatic $G_{12/13}$ signaling plays a physiological role in regulating blood glucose levels under certain nutritional conditions. These new data have been incorporated into the revised manuscript (Results, page 24, top paragraph; Discussion, page 29, bottom paragraph).

In this context, a previous report (Mutel *et al.*, 2011) is of particular interest. Mutel *et al.* generated mutant mice selectively lacking the *G δ pase* gene in hepatocytes (this gene is considered essential for HGP). Interestingly, these mutant mice showed unaltered fasting blood glucose levels, as compared to their control littermates. The authors went on to demonstrate that extrahepatic gluconeogenesis in the kidneys and intestine was able to maintain euglycemia. This study suggests that inactivation of a single hepatic pathway that can stimulate HGP can be easily compensated by an increase in extrahepatic gluconeogenesis.

Moreover, HGP is also regulated by the activity of the sympathetic nervous system, growth hormone, glucocorticoids, and several other factors acting directly on hepatocytes (Lin and Accili, 2011). It is possible that one or more of these factors contribute to the observation that blood glucose levels remained unaltered in HepG12/13 KO mice under most experimental conditions.

COMMENT

5- Why did the authors use hepatocyte specific TBG as marker for hepatocyte to generate the genetic model? Why did not they use albumin like commonly done?

OUR RESPONSE

Like albumin, TBG is selectively expressed in hepatocytes. Since we used the TBG promoter to achieve hepatocyte-specific expression of a G_i DREADD previously (Rossi *et al.*, 2018), we also relied on the use of the TBG promoter in the present study. Moreover, the AAV-TBG-Cre virus is widely used to achieve hepatocyte-specific deletion of floxed genes in mouse hepatocytes. We also used this virus successfully in the past to remove floxed STOP cassettes selectively in hepatocytes (Akhmedov *et al.*, 2017; Rossi *et al.*, 2018). The AAV-TBG-Cre virus is easily available through Addgene (<https://www.addgene.org/107787/>). The Addgene website lists 18 published papers that used this virus successfully. For all these reasons, we also used this virus in the present study.

COMMENT

6- It would be better to provide supportive immunohistochemistry images for the tissues to prove that the G12D is only expressed in liver and specifically in hepatocytes.

OUR RESPONSE

To address this issue, we separated mouse hepatocytes from other liver cells (Kupffer cells, stellate cells, etc.). Western blotting studies confirmed that the G12D designer receptor was expressed only in hepatocytes and not in other cell types of the liver prepared from Hep-G12D mice (Supplementary Fig. 3) (new).

To determine which percentage of hepatocytes expressed the G12D receptor, we used flow cytometry analysis to detect the presence of the HA epitope tag present at the extracellular N-terminus of G12D (Fig. 1a) (see Methods for details). This analysis showed that 55 ±8% of purified hepatocytes prepared from Hep-G12D mice expressed the G12D construct (mean ± s.e.m.; n=3). In contrast, no significant HA signal was detected with hepatocytes derived from control littermates (n=3).

Appropriate changes have been made at the beginning of the Results section in the revised manuscript (page 8 bottom, and page 9, top).

COMMENT

7- Authors showed in fig.1b western blot data about the specific expression of G12 in liver, do the authors have western blot or histology data for the Ga13 in the established model in the different tissues? Because they also assume that Ga13 is knocked down in their model.

OUR RESPONSE

Please note that Fig. 1b shows that the G12D receptor (not the G α_{12} G protein) is selectively expressed in the liver of LSL-G12D mice following i.v. injection with the AAV-TBG-Cre virus

(Hep-G12D mice) (please see the figure legend). In Fig. 3c, we show that AAV-TBG-Cre treatment of mice harboring the floxed *Gna13* allele (encoded protein: $G\alpha_{13}$) effectively knocks down the expression of $G\alpha_{13}$.

COMMENT

8- In fig.1F, why is the baseline blood glucose high even though the insulin tolerance test is performed after fasting.

OUR RESPONSE

For the ITT experiment, mice were fasted for only 4 hr. We now state this fact in the figure legend. This rather short fasting period explains why baseline blood glucose was relatively high in Fig. 1f.

COMMENT

9- Fig.3b, why is not there a *Gna12*^{-/-} *Gna13*^{fl/fl} with TBG-eGFP and TBG-Cre to know the role of *Ga13* alone.

OUR RESPONSE

We feel that the data shown in Fig. 3b allow us to draw conclusions regarding the role of $G\alpha_{13}$ alone. Please compare the red and blue curves in Fig. 3b. The mice represented by these curves are *Gna12*^{-/-} *Gna13*^{fl/fl} mutant mice expressing the G12D designer receptor in hepatocytes (mice were injected with AAV-TBG-G12D). These mice lack $G\alpha_{12}$ throughout the body and harbor a floxed *Gna13* allele (encoded protein: $G\alpha_{13}$; note that the floxed *Gna13* allele is fully functional (work done in the lab of Dr. Stefan Offermanns)). The mice represented by the red curve were co-injected with a pharmacological inert AAV (AAV-TBG-eGFP). CNO treatment of these mice resulted in a robust increase in blood glucose levels. The mice represented by the blue curve were co-injected with AAV-TBG-Cre, resulting in the loss of $G\alpha_{13}$ in hepatocytes (Fig. 3c). CNO treatment of these mice caused considerably smaller elevations in blood glucose levels that were not significantly different from those observed with WT mice lacking the G12D receptor (black curve) (Fig. 3b). These data allow us to conclude that hepatic $G\alpha_{13}$ makes a significant contribution to the G12D-mediated hyperglycemic effects observed in this in vivo assay.

In the text, we summarize these observations as follows (page 13, top): "This finding clearly indicates that the G12D-mediated hyperglycemic responses require the activation of both hepatic G_{12} and G_{13} and that no other classes of heterotrimeric G proteins are involved in this effect."

Also see the additional text in the Results section under "Both G_{12} and G_{13} are required for G12D-mediated hepatic glycogenolysis and gluconeogenesis" (pages 13 and 14 in the revised manuscript).

COMMENT

10- Fig.c7, the authors suppose in the graphical abstract that JNK is a downstream of ROCK, but they did not provide experimental evidence for that.

OUR RESPONSE

In Fig. 6c, we treated primary hepatocytes obtained from Hep-G12D mice with CNO or a mixture of CNO and Y-27632, a selective ROCK inhibitor (10 μ M each drug). Note that CNO stimulated the phosphorylation of JNK and that this effect was absent in the presence of Y-

27632. In addition, several previous studies have shown that activated ROCK can promote the formation of pJNK (Schofield and Bernard, 2013). These findings strongly suggest that JNK acts downstream of ROCK, as shown in Fig. 7c.

Reviewer #3 (Remarks to the Author):

This is an interesting and solid paper providing evidence that G12/13 signaling regulates hepatic glucose homeostasis.

COMMENT

Specific Concerns: 1. Given that the chemogenetic construct is modified it will be important to provide negative data for coupling to Gq/i/s-family G proteins (e.g. via BRET).

OUR RESPONSE

The G12D construct used in the present study has been characterized in a published article (Ono et al., 2023). This is stated in the first paragraph of the Results section. Using a nanobit-G-protein dissociation assay, Ono et al. demonstrated that the G12D designer receptor used in the present study showed little or no coupling to G_s, G_q-, or G_i-type G proteins.

To further confirm these findings in functional assays, we expressed the G12D construct in HEK293 cells (Supplementary Fig. 2) (new). We then treated the G12D-expressing cells with CNO (1 and/or 10 μM). We found that CNO did not affect intracellular IP₁ and cAMP levels. In contrast, CNO treatment of HEK293 cells expressing a G_{q/11}-coupled DREADD (hM3Dq) resulted in a robust increase in IP₁ production. Similarly, forskolin treatment of G12D-expressing HEK293 cells led to pronounced increases in cytoplasmic cAMP levels. Taken together, these data strongly support the concept that the G12D DREADD used in the present study selectively activates G proteins of the G₁₂ family.

These new functional data are shown in Supplementary Fig. 2. We also made appropriate changes in the revised manuscript (page 7, penultimate paragraph).

Also, please note: In our manuscript, we repeatedly show that the G12D-mediated metabolic effects are abolished or greatly reduced in the absence of Gα₁₂ and/or Gα₁₃, clearly indicating that the observed metabolic phenotypes required G12D-mediated activation of G_{12/13} signaling.

References

- Akhmedov, D., Mendoza-Rodriguez, M.G., Rajendran, K., Rossi, M., Wess, J., and Berdeaux, R. (2017). G_s-DREADD knock-in mice for tissue-specific, temporal stimulation of cyclic AMP signaling. *Mol Cell Biol* 37. 10.1128/mcb.00584-16.
- Avet, C., Mancini, A., Breton, B., Le Gouill, C., Hauser, A.S., Normand, C., Kobayashi, H., Gross, F., Hogue, M., Lukasheva, V., et al. (2022). Effector membrane translocation biosensors reveal G protein and βarrestin coupling profiles of 100 therapeutically relevant GPCRs. *Elife* 11. 10.7554/eLife.74101.
- Barroso, E., Jurado-Aguilar, J., Wahli, W., Palomer, X., and Vázquez-Carrera, M. (2024). Increased hepatic gluconeogenesis and type 2 diabetes mellitus. *Trends Endo Metab: TEM*. 10.1016/j.tem.2024.05.006.

- Bogoyevitch, M.A., and Kobe, B. (2006). Uses for JNK: the many and varied substrates of the c-Jun N-terminal kinases. *Microbiol Mol Biol Rev* 70, 1061-1095.
- Burgess, S.C. (2015). Regulation of glucose metabolism in liver. *International Textbook of Diabetes Mellitus*, 193-210.
- Deng, L., Shoji, I., Ogawa, W., Kaneda, S., Soga, T., Jiang, D.P., Ide, Y.H., and Hotta, H. (2011). Hepatitis C virus infection promotes hepatic gluconeogenesis through an NS5A-mediated, FoxO1-dependent pathway. *J Virol* 85, 8556-8568. 10.1128/jvi.00146-11.
- Exton, J.H. (1987). Mechanisms of hormonal regulation of hepatic glucose metabolism. *Diabet Metab Rev* 3, 163-183.
- Exton, J.H., Mallette, L.E., Jefferson, L.S., Wong, E.H., Friedmann, N., Miller, T.B., Jr., and Park, C.R. (1970). The hormonal control of hepatic gluconeogenesis. *Rec Progr Horm Res* 26, 411-461. 10.1016/b978-0-12-571126-5.50014-5.
- Hirosumi, J., Tuncman, G., Chang, L., Görgün, C.Z., Uysal, K.T., Maeda, K., Karin, M., and Hotamisligil, G.S. (2002). A central role for JNK in obesity and insulin resistance. *Nature* 420, 333-336. 10.1038/nature01137.
- Jiang, L.I., Collins, J., Davis, R., Fraser, I.D., and Sternweis, P.C. (2008). Regulation of cAMP responses by the G12/13 pathway converges on adenylyl cyclase VII. *J Biol Chem* 283, 23429-23439. 10.1074/jbc.M803281200.
- Jiang, L.I., Wang, J.E., and Sternweis, P.C. (2013). Regions on adenylyl cyclase VII required for selective regulation by the G13 pathway. *Mol Pharmacol* 83, 587-593. 10.1124/mol.112.082446.
- Li, J.H., Jain, S., McMillin, S.M., Cui, Y., Gautam, D., Sakamoto, W., Lu, H., Jou, W., McGuinness, O.P., Gavrilova, O., and Wess, J. (2013). A novel experimental strategy to assess the metabolic effects of selective activation of a G(q)-coupled receptor in hepatocytes in vivo. *Endocrinology* 154, 3539-3551. 10.1210/en.2012-2127.
- Lin, H.V., and Accili, D. (2011). Hormonal regulation of hepatic glucose production in health and disease. *Cell Metab* 14, 9-19. 10.1016/j.cmet.2011.06.003.
- Marinissen, M.J., Chiariello, M., Tanos, T., Bernard, O., Narumiya, S., and Gutkind, J.S. (2004). The small GTP-binding protein RhoA regulates c-jun by a ROCK-JNK signaling axis. *Mol Cell* 14, 29-41. 10.1016/s1097-2765(04)00153-4.
- Mutel, E., Gautier-Stein, A., Abdul-Wahed, A., Amigó-Correig, M., Zitoun, C., Stefanutti, A., Houberdon, I., Tourette, J.A., Mithieux, G., and Rajas, F. (2011). Control of blood glucose in the absence of hepatic glucose production during prolonged fasting in mice: induction of renal and intestinal gluconeogenesis by glucagon. *Diabetes* 60, 3121-3131. 10.2337/db11-0571.
- Ongusaha, P.P., Qi, H.H., Raj, L., Kim, Y.B., Aaronson, S.A., Davis, R.J., Shi, Y., Liao, J.K., and Lee, S.W. (2008). Identification of ROCK1 as an upstream activator of the JIP-3 to JNK signaling axis in response to UVB damage. *Science Sign* 1, ra14. 10.1126/scisignal.1161938.
- Ono, Y., Kawakami, K., Nakamura, G., Ishida, S., Aoki, J., and Inoue, A. (2023). Generation of Gai knock-out HEK293 cells illuminates Gai-coupling diversity of GPCRs. *Commun Biol* 6, 112.
- Piali, L., Froidevaux, S., Hess, P., Nayler, O., Bolli, M.H., Schlosser, E., Kohl, C., Steiner, B., and Clozel, M. (2011). The selective sphingosine 1-phosphate receptor 1 agonist ponesimod protects against lymphocyte-mediated tissue inflammation. *J Pharmacol Exp Ther* 337, 547-556. 10.1124/jpet.110.176487.
- Regard, J.B., Sato, I.T., and Coughlin, S.R. (2008). Anatomical profiling of G protein-coupled receptor expression. *Cell* 135, 561-571. 10.1016/j.cell.2008.08.040.

- Rossi, M., Zhu, L., McMillin, S.M., Pydi, S.P., Jain, S., Wang, L., Cui, Y., Lee, R.J., Cohen, A.H., Kaneto, H., et al. (2018). Hepatic Gi signaling regulates whole-body glucose homeostasis. *J Clin Invest* *128*, 746-759. 10.1172/jci94505.
- Schofield, A.V., and Bernard, O. (2013). Rho-associated coiled-coil kinase (ROCK) signaling and disease. *Crit Rev Biochem Mol Biol* *48*, 301-316. 10.3109/10409238.2013.786671.
- Wang, M.C., Bohmann, D., and Jasper, H. (2005). JNK extends life span and limits growth by antagonizing cellular and organism-wide responses to insulin signaling. *Cell* *121*, 115-125. 10.1016/j.cell.2005.02.030.
- Weng, Q., Liu, Z., Li, B., Liu, K., Wu, W., and Liu, H. (2016). Oxidative stress induces mouse follicular granulosa cells apoptosis via JNK/FoxO1 pathway. *PLoS One* *11*, e0167869. 10.1371/journal.pone.0167869.
- Zeke, A., Misheva, M., Reményi, A., and Bogoyevitch, M.A. (2016). JNK signaling: regulation and functions based on complex protein-protein partnerships. *Microbiol Mol Biol Rev* *80*, 793-835. 10.1128/mnbr.00043-14.